# AD-MIR: Bridging the Gap from Perception to Persuasion in Advertising Video Understanding via Structured Reasoning

**Binxiao Xu** [1 2 *]  **Junyu Feng** [2 *]  **Xiaopeng Lin** [2]  **Haodong Li** [3]  **Zhiyuan Feng** [4]  **Bohan Zeng** [1]
**Ruichuan An** [1 †]  **Ming Lu** [5]  **Qi She** [6]  **Wentao Zhang** [1 ‡]

## Abstract

Multimodal understanding of advertising videos is essential for interpreting the intricate relationship between visual storytelling and abstract persuasion strategies. However, despite excelling at general search, existing agents often struggle to bridge the cognitive gap between pixel-level perception and high-level marketing logic. To address this challenge, we introduce **AD-MIR**, a framework designed to decode advertising intent via a two-stage architecture. First, in the **Structure-Aware Memory Construction** phase, the system converts raw video into a structured database by integrating semantic retrieval with exact keyword matching. This approach prioritizes fine-grained brand details (e.g., logos, on-screen text) while dynamically filtering out irrelevant background noise to isolate key protagonists. Second, the **Structured Reasoning Agent** mimics a marketing expert through an iterative inquiry loop, decomposing the narrative to deduce implicit persuasion tactics. Crucially, it employs an evidence-based self-correction mechanism that rigorously validates these insights against specific video frames, automatically backtracking when visual support is lacking. Evaluation on the AdsQA benchmark demonstrates that AD-MIR achieves state-of-the-art performance, surpassing the strongest general-purpose agent, DVD, by 1.8 and 9.5 percentage points in strict and relaxed accuracy, respectively. These results underscore that effective advertising understanding demands explicitly grounding abstract mar-

keting strategies in pixel-level evidence. The code is available at https://github.com/Little-Fridge/AD-MIR.

## 1. Introduction

The research paradigm for video understanding is shifting from entity-centric perception to intent-oriented cognition, as Lin et al. (2024); Zhang et al. (2025); Fu et al. (2025) progressively move from object-centric recognition to higher-level reasoning over long-form video content. Despite the success of Large Multimodal Models (LMMs) in generalizing across objective physical descriptions, a substantial cognitive disparity remains when processing subjective and strategic content, even as Liu et al. (2023); Zhang et al. (2024a) demonstrate strong generalization under instruction-tuned and long-context settings. Advertising videos serve as an adversarial benchmark for this limitation. Unlike objective recordings, ads are engineered semiotic systems (Hussain et al., 2017). In such non-linear narratives, cinematic techniques such as lighting manipulation and rhythmic editing function not merely as visual signals, but as vehicles for abstract persuasion logic. Therefore, bridging this semantic divide between low-level visual facts (e.g., a scene fading to black) and high-level abstract intents (e.g., creating a sense of suppression to trigger pain points) is essential for effective advertising understanding.

*Table 1.* **Feature comparison with existing paradigms.** This high-level capability taxonomy follows the reported design goals of each method: general agents (e.g., DVD) primarily emphasize retrieval, RL-optimized baselines (ReAd-R) optimize QA alignment implicitly, whereas AD-MIR explicitly integrates expert modules for persuasion decoding ("why") and visual-anchor verification for hallucination control.

| Method | Perception | | Strategy Cognition | | Reliability |
| --- | --- | --- | --- | --- | --- |
| | Visual Grounding | Fine-grained Detail | Causal Reasoning | Persuasion Decoding | Visual Verification |
| End-to-End LMMs | ✗ | ✓ | ✗ | ✗ | ✗ |
| General Video Agents (DVD) | ✓ | ✓ | ✗ | ✗ | ✗ |
| ReAd-R (AdsQA Baseline) | ✓ | ✓ | ✓ | ✗ | ✗ |
| **AD-MIR (Ours)** | ✓ | ✓ | ✓ | ✓ | ✓ |

Current video understanding models face significant hurdles in the domain of advertising (Hussain et al., 2017). While

---
[*]Equal contribution [†] Project leader [‡] Corresponding author [1]Peking University, Beijing, China [2]Xi'an Jiaotong University, Xi'an, China [3]South China University of Technology, Guangzhou, China [4]Tsinghua University, Beijing, China [5]Intel, Beijing, China [6]ByteDance, Beijing, China. Correspondence to: Wentao Zhang <wentao.zhang@pku.edu.cn>.

*Proceedings of the 43rd International Conference on Machine Learning*, Seoul, South Korea. PMLR 306, 2026. Copyright 2026 by the author(s).

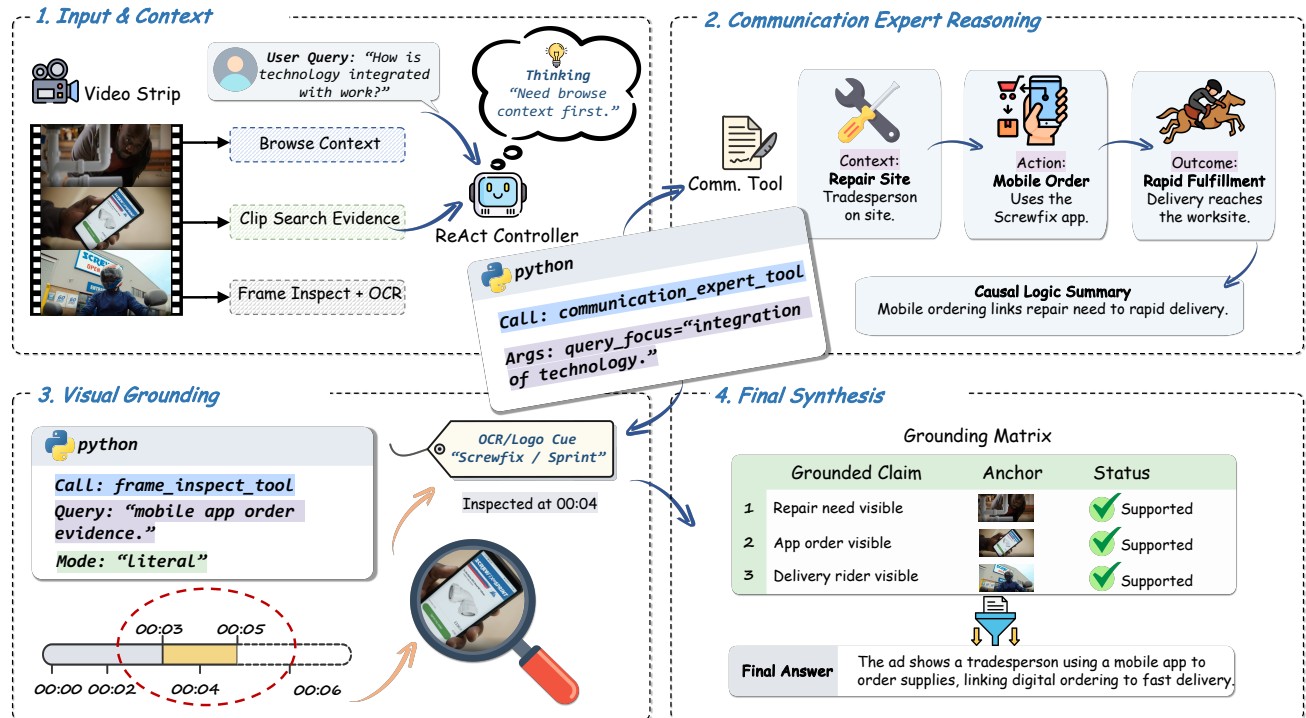

*Figure 1.* **Illustrative walkthrough of AD-MIR's reasoning pipeline on an AdsQA example.** Unlike general agents that focus on retrieval, AD-MIR bridges the cognitive gap by first constructing browse-based context and a high-level causal narrative via a communication expert (Phase 1–2), then performing targeted frame inspection to ground precise visual details (Phase 3), and finally synthesizing an answer supported by concrete visual anchors (Phase 4).

end-to-end LMMs excel at describing general visual content (Liu et al., 2023; Lin et al., 2024), they struggle with two main limitations: limited memory and logic hallucinations (Zhang et al., 2024a; Guan et al., 2024; Fu et al., 2025). First, constrained by context windows, these models often lose track of details across long narratives. Second, and more critically, lacking a mechanism to decode persuasive intent, they tend to hallucinate when explaining strategic motivations, often fabricating reasons that defy basic advertising logic (Hussain et al., 2017; Guan et al., 2024).

On the other hand, tool-augmented agents, represented by the Deep Video Discovery (DVD) Agent (Zhang et al., 2025), have successfully established a baseline for *visual grounding* by precisely locking onto the timestamps of events. However, a critical distinction exists between retrieving *"what happened"* and reasoning *"why it happened."* While DVD demonstrates acuity in locating visual content, it lacks the cognitive capability to decode the underlying marketing psychology(e.g., identifying rhetorical devices, emotional appeals, and symbolic metaphors). Even the recent ReAd-R framework (Long et al., 2025), which employs reinforcement learning to enhance retrieval-QA alignment, remains limited by implicit data fitting. Without explicit structural constraints for persuasion analysis, such models

struggle to distinguish between superficial visual correlations and deep strategic causation (e.g., recognizing a phone not merely as an object, but as a solution to a labor problem). Consequently, when nuanced strategic inquiries arise, standard video agents and RL-optimized baselines can only draw shallow associations, resulting in a logical fracture between visual perception and intent understanding. Table 1 summarizes the capability gaps between these existing paradigms and our proposed approach.

In response to these challenges, we propose **AD-MIR** (**AD**vertising **M**ultimodal **I**ntent **R**easoning), a multimodal agent framework tailored for advertising videos. Rather than relying on single-shot retrieval or black-box inference, AD-MIR is designed to close the cognitive gap via a two-stage architecture that extends the standard reasoning-and-acting paradigm (Yao et al., 2022). As illustrated in Figure 1, the framework operates by mirroring how a human analyst works: first, constructing browse-based narrative context from multimodal evidence, and second, employing an iterative inquiry loop to align high-level hypotheses with precise visual anchors.

The first phase is *Structure-Aware Memory Construction*, where the system decouples raw video streams into visual,

textual, and audio modalities to construct a hierarchical index. This allows the model to anchor cues often missed by general retrieval. The second phase involves the *Structured Reasoning Agent*, which employs an explicit iterative inquiry-refinement mechanism. Instead of a generic loop, this mechanism intelligently selects the most suitable tools based on the query's nature. For factual queries, it routes to perception tools; for strategic queries, it invokes specialized communication expert modules grounded in marketing psychology (e.g., analyzing persuasion, rhetoric, and emotional arcs). This hierarchical synergy enables AD-MIR to capture implicit logical associations, such as linking "a plumber under a sink" to "a speeding motorbike" by reconstructing discrete visual symbols into a coherent marketing narrative.

Our main contributions are summarized as follows:

1. **A Multimodal Reasoning Agent Framework Tailored for Advertising.** We present AD-MIR, a novel agent designed to bridge the gap between visual perception and intent understanding. By employing an iterative inquiry-refinement mechanism that aligns expert-derived marketing insights with rigorous visual-anchor verification, it effectively decodes complex persuasion tactics that general agents miss.

2. **Hierarchical Perception-Inference Toolchain.** We introduce a query-driven architecture that selectively routes inquiries between perception tools for factual grounding and specialized expert modules grounded in marketing psychology. This specialized design enables the precise interpretation of symbolic imagery and emotional cues, surpassing the literal description capabilities of standard multimodal systems.

3. **Visual-Anchor Verification for Grounded Reasoning.** We introduce a mechanism that validates high-level reasoning against pixel-level evidence, ensuring interpretations are derived strictly from video frames rather than linguistic priors. This cross-modal alignment bridges abstract marketing concepts with concrete visual signals, significantly enhancing the reliability and interpretability of persuasion analysis.

Related work is summarized in Appendix A.

## 2. Method

In this section, we present the AD-MIR framework. As illustrated in Figure 2, AD-MIR utilizes an agentic reasoning architecture that couples a ReAct-style controller with a shared multimodal database $\mathcal{M}$ and a suite of domain-specific interaction tools. This design effectively bridges the semantic gap between pixel-level perception and high-level marketing intent. The workflow proceeds in two synergistic stages: (1) **Structure-Aware Memory Construction**,

where raw video streams are decoupled into a discrete, indexable multimodal database to isolate key narrative elements; and (2) **Iterative Intent Reasoning**, where a domain-adaptive agent orchestrates perception primitives and intent experts via a "Think-Act-Observe" loop, aligning visual evidence with logically grounded marketing inference. The final output module then verifies concrete anchors and compresses the answer without dropping entities, numbers, attributes, or negation.

### 2.1. Problem Definition

We formalize the advertising video understanding task as a Partially Observable Markov Decision Process (POMDP), defined by the tuple $(\mathcal{S}, \mathcal{A}, \mathcal{T}, \mathcal{R}, \Omega, \mathcal{O})$ (Wu et al., 2019). Due to the non-linear and metaphorical nature of advertising narratives, the agent cannot perceive the full video $\mathcal{V}$ in a single step, rendering the state space $\mathcal{S}$ partially observable. In our setting, $\mathcal{A}$ denotes the finite set of tool calls in Table 2, $\Omega$ denotes textual or visual observations returned by these tools, $\mathcal{T}$ is the deterministic context update induced by appending observations to the interaction history, and $\mathcal{R}$ is an implicit terminal utility represented by answer correctness and visual support rather than a learned reward model. This formulation is used to specify the inference process and action constraints; AD-MIR does not train a POMDP policy.

**Definition 2.1.** The state $s_t \in \mathcal{S}$ at time $t$ is defined as the union of the multimodal database $\mathcal{M}$, the context-anchored subject registry $\mathcal{S}_{reg}$, and the interaction history $\mathcal{H}_{t-1}$.

Given a natural language query $Q$, the agent receives an environmental observation $o_t \in \Omega$. It then generates a Chain-of-Thought (CoT) reasoning step $z_t$ by leveraging the history $\mathcal{H}_{t-1} = \{o_0, a_0, \ldots, a_{t-1}, z_{t-1}\}$.

**Proposition 2.2.** *The agent selects the next action $a_t \in \mathcal{A}$ according to a domain-adaptive policy $\pi(a_t|\mathcal{H}_{t-1}, o_t; \Theta)$, where $\Theta$ represents the frozen parameters of a Large Multimodal Model. The policy is implemented via prompt-based in-context learning rather than gradient-based updates.*

As a conceptual objective for inference-time reasoning, the system prioritizes an answer $A^*$ that is semantically consistent with the query and grounded in retrieved evidence:

$$A^* = \arg\max_A P(A|\mathcal{V}, Q, \mathcal{S}_{reg}; \Theta) \qquad (1)$$

### 2.2. Structure-Aware Memory Construction

To mitigate the context and computational bottlenecks caused by dense multimodal advertising narratives and simultaneously ensure granular evidence backtracking, we employ a dedicated offline pipeline to decouple the continuous raw video stream into a structured multimodal database.

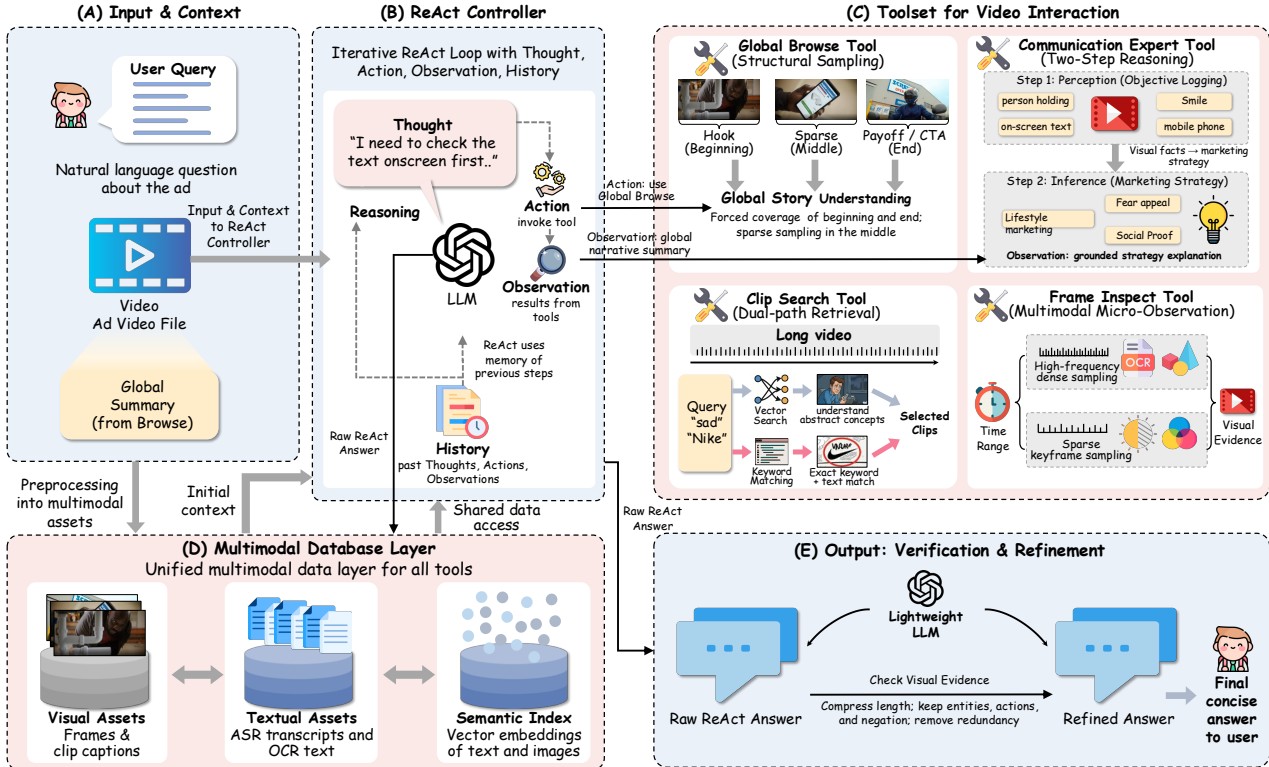

*Figure 2.* **The overall architecture of AD-MIR.** The framework comprises five synergistic components: (A) Input & Context construction for multi-modal preprocessing; (B) a ReAct Controller for iterative reasoning; (C) a Hierarchical Toolset featuring global browsing, communication experts, and fine-grained inspection; (D) a Unified Multimodal Database serving as shared memory; and (E) an Output Verification and Refinement stage to ensure concise, evidence-based final answers.

This systematic preprocessing workflow is primarily underpinned by two pivotal mechanisms:

### 2.2.1. HYBRID SEMANTIC-LEXICAL INDEXING

We discretize the raw video stream into a set of temporal clips $\mathcal{C} = \{c_i\}_{i=1}^N$. To construct a comprehensive semantic index, for each clip $c_i$, we utilize a VLM to generate fine-grained visual narrative descriptions $d_{\text{vis}}^{(i)}$ and integrate the Whisper model to extract timestamped audio transcripts $d_{\text{aud}}^{(i)}$. Addressing the precision loss inherent in single dense vector retrieval when handling specific entities (e.g., brand names, prices), we propose a hybrid semantic-lexical scoring strategy. Given a query $Q$, the relevance score $S(Q, c_i)$ for clip $c_i$ is formalized as a weighted linear combination of dense semantic similarity and exact keyword matching:

$$S(Q, c_i) = \underbrace{\cos(\phi(Q), \phi(d_{\text{vis}}^{(i)} \oplus d_{\text{aud}}^{(i)}))}_{\text{Semantic}} + \beta \cdot \underbrace{\mathcal{K}(Q, d_{\text{vis}}^{(i)} \oplus d_{\text{aud}}^{(i)})}_{\text{Lexical}} \quad (2)$$

where $\phi(\cdot)$ denotes the semantic text embedding function, and $\oplus$ represents the concatenation operation of multimodal text streams. $\mathcal{K}(\cdot)$ is a lexical scoring function based on keyword hit rate and exact match enhancement, and $\beta$ is a

hyperparameter regulating the relative lexical weight. This formulation ensures that the agent possesses dual perception capabilities: the dense vector component captures abstract visual concepts (e.g., "a warm dinner"), while the sparse retrieval component precisely localizes specific explicit marketing clues (e.g., "Nike" or "50% OFF").

### 2.2.2. CONTEXT-ANCHORED SUBJECT REGISTRY

In complex advertising scenarios, the frame is often crowded with background characters, which distracts the agent from the main protagonists. To mitigate this noise, we implement a two-step register-and-filter mechanism.

First, during the offline memory construction, we use the selected captioning VLM to extract a static subject registry $\mathcal{S}_{reg} = \{s_j\}_{j=1}^M$. Unlike simple object detection, the model generates a rich semantic profile $d_j$ for each distinct character (e.g., "A middle-aged doctor wearing a stethoscope and looking anxious"), rather than just a bounding box label.

Second, during inference, we apply a dynamic activation strategy to decide which characters are relevant. User queries often suffer from referential ambiguity (e.g., "What

*Table 2.* **Action Spaces for Intent Reasoning.** The framework stratifies the action space into perceptual grounding tools and intent analysis experts to bridge the semantic gap. Specific implementation parameters (e.g., sampling rates) are detailed in Appendix B.2.

| Action | Function & Parameters |
|---|---|
| GLOBAL BROWSE | **Input:** Database $\mathcal{M}$, Query $Q_{init}$
Retrieves pre-computed narrative logs to synthesize a structured global summary $C_{glob}$. |
| CLIP SEARCH | **Input:** Database $\mathcal{M}$, Query $\hat{Q}$
Performs hybrid retrieval with Temporal Fusion to merge continuous event segments. |
| FRAME INSPECT | **Input:** Query $Q_{vis}$, Range $[t_s, t_e]$, Mode $m$
Variable-density visual inspection: $m_{lit}$ (dense) for factual details; $m_{sem}$ (sparse) for stylistic elements. |
| COMM. EXPERT | **Input:** Focus $\hat{Q}_{foc}$, Range $[t_s, t_e]$
Executes high-density grid inference to resolve long-term dependencies and decode persuasion logic. |
| FINISH | **Input:** Response $A$, Evidence $\mathcal{E}$
Triggers VERIFYGROUNDING; returns $A^*$ if verified, otherwise initiates a forced backtrack. |

is *he* holding?"), where the pronoun's target is unclear without context. To resolve this, we construct an enhanced semantic anchor $A_{anchor}$—a composite representation that fuses the raw query $Q$ with the global narrative summary $C_{glob}$ obtained from the global browse tool ($A_{anchor} = Q \oplus C_{glob}$). This anchor serves as a contextualized reference point in the vector space, ensuring the search is driven by the full narrative intent rather than ambiguous keywords. We then compute the relevance score $\rho_j$ between this anchor and each character's profile:

$$\mathcal{S}_{active} = \text{TopK}\left(\{s_j \in \mathcal{S}_{reg} \mid \rho_j = \cos(\phi(d_j), \phi(A_{anchor}))\}, k\right) \quad (3)$$

By filtering based on semantic similarity $\rho_j$, this mechanism acts as a cognitive spotlight: it activates relevant characters (e.g., the "doctor") while suppressing background noise, ensuring the reasoning chain focuses solely on evidence pertinent to the narrative intent.

### 2.3. Structured Reasoning Agent

At the core of AD-MIR lies a controller designed to emulate the cognitive processes of marketing experts by dynamically orchestrating action primitives.

#### 2.3.1. PROMPT-GUIDED REACT CONTROLLER.

We instantiate the selected LMM backbone as the central controller $\pi_\Theta$. Distinct from fine-tuned video agents, our

---

**Algorithm 1** AD-MIR Inference Process

**Input:** Video Stream $\mathcal{V}$, Query $Q$
**Output:** Refined answer $A^*$
**Stage I: Structure-Aware Memory Construction**
$\mathcal{M}, \mathcal{S}_{reg} \leftarrow \text{BUILDDB}(\mathcal{V})$ {1 FPS, ASR, backbone VLM}
**Stage II: Iterative Intent Reasoning**
$C_{glob}, \text{ASR} \leftarrow \text{GLOBALBROWSE}(\mathcal{M}, Q)$
$\mathcal{H}_0 \leftarrow \{Q, C_{glob}, \mathcal{S}_{reg}\}$
$t \leftarrow 0$
**while** $t < T_{\max}$ **do**
  $z_t, a_t \leftarrow \pi_\Theta(\mathcal{H}_t)$
  **if** $a_t = \text{FINISH}(A, \mathcal{E})$ **then**
    $\mathcal{W}_{anchors} \leftarrow \text{EXTRACTENTITIES}(\mathcal{E})$
    **if** $\text{VERIFYGROUNDING}(A, \mathcal{E}, \mathcal{W}_{anchors})$ **then**
      **return** $\text{VISUALANCHORREFINE}(A)$
    **end if**
    $\mathcal{H}_{t+1} \leftarrow \mathcal{H}_t \cup \{\text{"Reject: Weak Evidence"}\}$
    $t \leftarrow t + 1$
    **continue**
  **end if**
  **if** $a_t \in \mathcal{A}_{percept}$ **then**
    $o_t \leftarrow \text{EXECUTE}(a_t, \mathcal{M})$
  **else**
    **if** $a_t = \text{COMMEXPERT}(\hat{Q}_{foc}, [t_s, t_e])$ **then**
      $L_{vis} \leftarrow \text{GRIDSAMPLE}(\mathcal{M}, [t_s, t_e])$
      $o_t \leftarrow \text{REASON}(L_{vis}, \text{ASR}, C_{glob}, \hat{Q}_{foc})$
    **end if**
  **end if**
  $\mathcal{H}_{t+1} \leftarrow \mathcal{H}_t \cup \{(z_t, a_t, o_t)\}$
  $t \leftarrow t + 1$
**end while**
**return** FAILURE

---

system utilizes frozen parameters $\Theta$ governed by structured in-context learning. Instead of relying on implicit priors, the system prompt injects explicit behavioral constraints, mandating that the agent prioritizes retrieved visual evidence over internal parametric knowledge when interpreting abstract concepts. The execution logic follows the "Think-Act-Observe" cycle formalized in Algorithm 1.

At each step $t$, the agent generates a thought $z_t$ and selects an action $a_t$ conditioned on the interaction history $\mathcal{H}_{t-1}$. The controller is equipped with a self-correction mechanism: if the generated answer fails the visual grounding verification check (as detailed in Section 2.4), the system triggers a forced trajectory backtrack, appending a "Weak Evidence" constraint to $\mathcal{H}_t$ to guide the policy toward re-retrieval.

#### 2.3.2. SEARCH-CENTRIC TOOLSET CONSTRUCTION.

Building upon the structured database $\mathcal{M}$, we categorize the action space $\mathcal{A}$ into two distinct groups: *information*

*retrieval tools* for locating visual evidence and *reasoning modules* for decoding narrative logic. To ensure deterministic interaction between the agent and the environment, we define explicit functional interfaces for each tool, as comprehensively formalized in Table 2. The implementation details of these tools are elaborated below.

**Tool: Global Browse.** Serving as the strategic initializer for the reasoning trajectory, this tool constructs a holistic narrative framework $C_{glob}$ to ground subsequent actions. In contrast to conventional methods relying on stochastic frame sampling that frequently fractures temporal continuity, this module exploits the pre-computed textual modalities within $\mathcal{M}$. Specifically, it retrieves the top $K$ semantically relevant clip captions together with aligned ASR snippets, orders them chronologically, and feeds this compact evidence stream into the LMM. The resulting structured summary classifies the macro-genre (e.g., emotional narrative vs. direct promotion) and extracts pivotal entities. This global prior effectively prunes the search space by imposing essential semantic constraints, thereby preventing the agent from hallucinating in unrelated contexts.

**Tool: Clip Search.** To secure narrative continuity at the event level, this tool facilitates medium-granularity retrieval, addressing the issue of temporal fragmentation common in standard top-k retrieval. By executing a synthetic query $\hat{Q}$ against the hybrid index, it identifies candidate intervals which are subsequently refined via a Temporal Fusion Algorithm. Specifically, disjointed clips exhibiting minimal temporal gaps (less than 3 seconds) or high semantic affinity ($> 0.8$) are coalesced into unified event blocks. This mechanism effectively reconstructs fragmented retrieval results into continuous event units, ensuring that the agent perceives complete long-duration actions rather than isolated snippets.

**Tool: Frame Inspect.** Functioning as the system's visual microscope, this tool bridges the granularity gap between abstract semantics and raw pixel data by modulating its sampling strategy according to the analysis mode $m$. Specifically, the literal mode ($m_{lit}$) enforces high-frequency dense sampling to rigorously extract observable axioms, such as OCR text and object counts, which are essential for factual verification. In contrast, the semantic mode ($m_{sem}$) adopts a sparse keyframe strategy to interpret stylistic elements like lighting, color temperature, and metaphorical imagery. This dual-path architecture allows the agent to dynamically alternate between forensic precision and aesthetic appreciation based on the specific requirements of the query.

**Tool: Communication Expert.** Designed to bridge the gap between concrete visual signals and abstract persuasive intent, this tool introduces a Spatio-Temporal Grid Projection mechanism. Unlike conventional sequential processing that can lose cross-scene dependencies, this module transforms the temporal dimension into a unified spatial representation

by arranging $N = 64$ sampled frames into high-resolution composite visual grids. This projection enables the LMM to reason over the narrative arc while preserving enough visual density for scene-level grounding. The expert is not a free-form oracle: it operationalizes marketing priors from classical advertising and persuasion theory, including attention-interest-desire-action staging, social influence principles, and central/peripheral cue analysis (Strong, 1925; Cialdini, 2001; Petty & Cacioppo, 1986). Concretely, it first identifies observable narrative evidence (characters, product reveal, slogan, conflict, payoff), then maps those observations to persuasive functions (e.g., fear appeal, social proof, expectation reversal). To mitigate hallucinations in abstract deduction, we implement a Text-Constrained Symbolic Mapping protocol. By fusing the visual grid with aligned ASR and global summaries, the agent grounds high-level interpretations in observable scenes and on-screen text rather than unsupported linguistic priors.

### 2.4. Reasoning Reliability Mechanisms

We embed two control mechanisms within the execution loop to enforce a robust and coherent reasoning trajectory.

**Macro-to-Micro Evidence Zooming.** We enforce a hypothesis-driven cascading strategy:

$$\mathcal{C}_{\text{glob}} \xrightarrow{\text{Init}} \text{Intent} \xrightarrow{\text{Search}} \mathcal{T}_{rel} \xrightarrow{\text{Verify}} \mathcal{L}_{\text{vis}}. \qquad (4)$$

The agent first pre-emptively acquires the narrative contour via GLOBAL BROWSE to initialize the semantic context. Subsequently, within the ReAct loop, it prioritizes the COMM. EXPERT to formulate high-level persuasion hypotheses (e.g., identifying a specific marketing strategy). These hypotheses guide the CLIP SEARCH to narrow down relevant temporal intervals $\mathcal{T}_{rel}$, and finally, the agent locks onto pixel-level evidence through FRAME INSPECT for rigorous verification. This zooming mechanism prevents disorientation by ensuring that low-level visual search is always directed by high-level semantic intent.

**Visual Grounding Verification.** To mitigate hallucinations, we introduce a rigorous answer verification protocol. Upon generating a candidate answer $A_{\text{cand}}$, the system triggers a grounding function $\Psi$:

$$\text{State} = \Psi(A_{\text{cand}}, \mathcal{E}, \mathcal{W}_{\text{anchors}}) \qquad (5)$$

where $\mathcal{E}$ denotes the accumulated evidence chain, and $\mathcal{W}_{\text{anchors}}$ represents the set of visual entities extracted from the retrieving history. The function $\Psi$ checks whether the key claims in $A_{\text{cand}}$ are supported by specific frames in $\mathcal{W}_{\text{anchors}}$. If visual grounding is insufficient (i.e., State = Reject), the system mandates a forced trajectory backtrack, appending a negative constraint to the history to guide the agent toward alternative retrieval paths. We use a source

*Table 3.* Performance comparison on the AdsQA benchmark. We report both strict and relaxed accuracy (%) across all five dimensions (VU, ER, TE, PS, AM) and the overall average. AD-MIR, built upon Qwen2.5-VL-7B and o1, is evaluated against commercial LMMs (o1), open-source video LVLMs, and reasoning-oriented video agents.

| Model | Strict Accuracy | | | | | | Relaxed Accuracy | | | | | |
|---|---|---|---|---|---|---|---|---|---|---|---|---|
| | VU | ER | TE | PS | AM | Overall | VU | ER | TE | PS | AM | Overall |
| *Commercial Large Multimodal Model* | | | | | | | | | | | | |
| o1(Jaech et al., 2024) | 31.8 | 32.0 | 34.9 | 31.0 | 32.0 | 33.1 | 50.9 | 52.8 | 54.8 | 50.0 | 51.2 | 52.4 |
| *Open-sourced Video-LLMs* | | | | | | | | | | | | |
| VideoLLaMA2-7B(Cheng et al., 2024) | 4.56 | 7.75 | 7.28 | 6.06 | 7.38 | 6.48 | 21.6 | 29.0 | 28.4 | 22.6 | 27.2 | 25.2 |
| LLaVA-OneVision-7B(Li et al., 2024) | 11.8 | 11.6 | 16.2 | 13.7 | 15.1 | 14.0 | 35.8 | 39.5 | 43.2 | 36.8 | 41.7 | 39.1 |
| LLaVA-Video-7B(Zhang et al., 2024b) | 14.0 | 14.4 | 18.7 | 15.2 | 17.3 | 16.1 | 37.8 | 41.6 | 45.1 | 38.1 | 43.6 | 41.0 |
| Qwen2.5-VL-7B(Bai et al., 2025) | 27.6 | 27.1 | 25.7 | 26.3 | 25.0 | 26.2 | 49.8 | 51.6 | 48.8 | 48.5 | 47.2 | 49.0 |
| *Reasoning Agent/Models* | | | | | | | | | | | | |
| ReAd-R(Qwen2.5-VL-7B)(Long et al., 2025) | 20.4 | 27.9 | 27.2 | 22.1 | 25.5 | 25.0 | 46.2 | 56.2 | 54.6 | 48.2 | 52.6 | 51.5 |
| DVD(o1)(Zhang et al., 2025) | 31.1 | 34.0 | 39.8 | 35.9 | 36.6 | 36.3 | 45.0 | 50.9 | 54.8 | 48.5 | 51.5 | 50.5 |
| AD-MIR(Qwen2.5-VL-7B) | 31.6 | 34.0 | 31.0 | 29.5 | 27.4 | 30.7 | 52.5 | 55.9 | 53.0 | 50.2 | 50.1 | 52.1 |
| AD-MIR(o1) | **32.8** | **36.1** | **41.6** | **37.9** | **38.1** | **38.1** ↑1.8 | **56.0** | **59.9** | **63.2** | **58.8** | **60.6** | **60.0** ↑9.5 |

hierarchy rather than blind over-verification. Concrete visual claims, such as object identity, OCR text, brand names, and temporal events, must be supported by CLIP SEARCH, FRAME INSPECT, or explicit grid evidence from COMM. EXPERT. Abstract persuasion labels are treated as hypotheses derived from those anchors; they are rejected when they lack supporting anchors or contradict inspected frames, but are not rejected merely because a sparse literal inspection cannot name the abstract strategy itself.

# 3. Experiments

In this section, we evaluate AD-MIR on the AdsQA benchmark (Long et al., 2025). Our experimental design aims to: (1) benchmark the framework against end-to-end LMMs and general-purpose video agents; (2) isolate the contributions of specific architectural components via ablation studies; and (3) analyze hyperparameter sensitivity to verify system robustness and reproducibility.

## 3.1. Experimental Setup

**Dataset.** We evaluate AD-MIR on AdsQA, which covers 38 advertising categories and asks questions ranging from factual retrieval to rhetorical strategy and audience modelling. This benchmark directly tests whether an agent can connect pixel-level evidence to high-level persuasive intent.

**Evaluation Metrics.** Following the official AdsQA protocol, an LMM judge scores each prediction against the video meta-data, query, and ground truth with $s \in \{0, 0.5, 1.0\}$; we report **strict accuracy** as the fraction of examples with $s = 1.0$ and **relaxed accuracy** as the official partial-credit average, where scores of 1.0, 0.5, and 0 contribute 1, 0.5, and 0 points, respectively. We compute both metrics over VU, ER, TE, PS, and AM. To avoid verbosity-based judge

artifacts, a format-only Refinement Module prompts the model to produce concise answers while preserving entities, attributes, numbers, and negation; when visual anchoring is required, it further prompts the answer to remain under 30 tokens. Prompts are provided in Appendices F and F.4.

**Baselines.** We compare against three baseline families in Table 3: commercial end-to-end LMMs, recent open-source video LVLMs, and reasoning-oriented video agents such as ReAd-R and DVD. AD-MIR is evaluated with both Qwen2.5-VL-7B and o1 backbones; implementation details appear in Appendix B.1.

## 3.2. Main Results

We present the comprehensive performance comparison in Table 3. All gains are reported in percentage points. As evidenced by the data, AD-MIR establishes new state-of-the-art benchmarks across both strict and relaxed accuracy metrics, yielding three critical insights into the efficacy of structured reasoning agents.

**Unlocking Latent Reasoning Capabilities.** Our results demonstrate substantial performance gains over base models across both metrics. Equipped with the o1 backbone, AD-MIR achieves a strict accuracy of **38.1%** (+5.0 over o1; +1.8 over DVD) and a relaxed accuracy of **60.0%** (+7.6 over o1; +9.5 over DVD). Similarly, AD-MIR (Qwen2.5-VL-7B) improves upon its base model by **+4.5** (strict) and **+3.1** (relaxed). These consistent improvements indicate that the bottleneck in complex video understanding lies less in inherent parametric knowledge and more in the orchestration of long-horizon reasoning. By enforcing an explicit "Think-Act-Observe" loop, AD-MIR effectively unlocks latent deductive capabilities, bridging the gap between raw knowledge and precise decision-making.

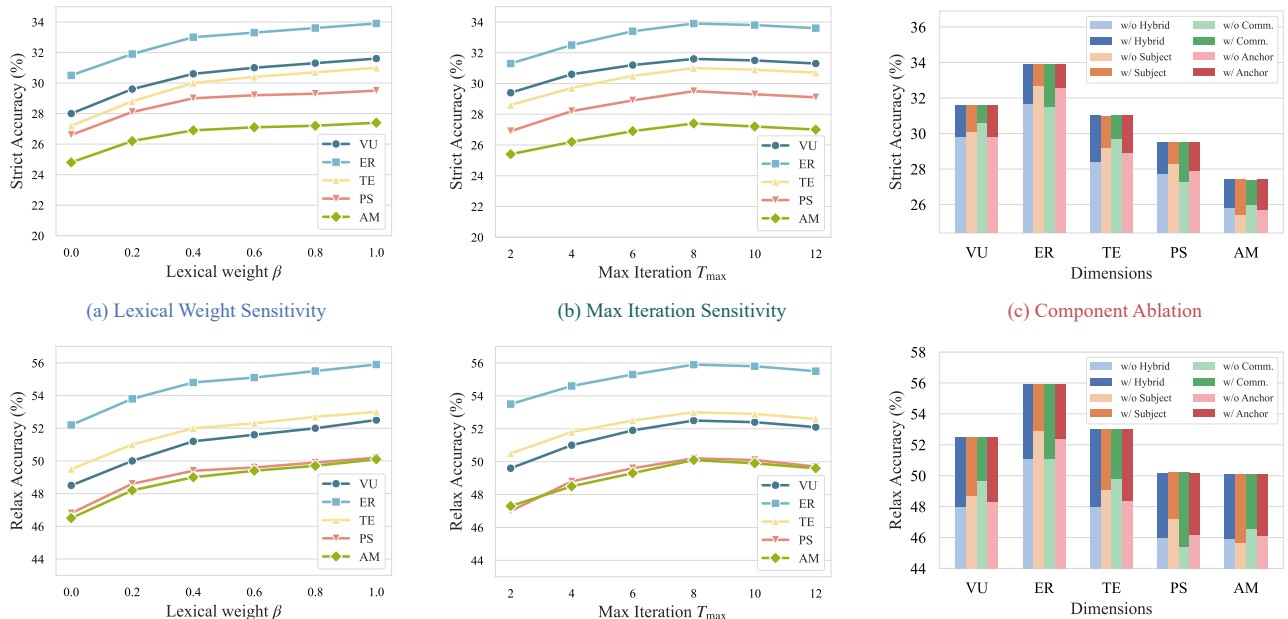

*Figure 3.* Ablation and sensitivity analysis of **AD-MIR** on AdsQA. Subfigures (a) and (b) report strict (top) and relaxed (bottom) accuracy on all five dimensions under different lexical weights $\beta$ and maximum reasoning steps $T_{max}$, respectively, showing that performance is stable in a broad range and peaks around the default setting. Subfigure (c) presents component-level ablations for Hybrid Indexing, Subject Registry, Communication Expert, and Visual Anchor, where "w/" and "w/o" denote whether the corresponding module is enabled; the results highlight the complementary gains of structured indexing, domain expert reasoning, and visual anchor self-correction.

**Bridging the Proprietary-Open Source Gap.** Crucially, our framework enables smaller open-source models to compete with larger commercial systems. AD-MIR (Qwen2.5-VL-7B) attains a relaxed accuracy of **52.1%**, comparable to the commercial o1 baseline (52.4%). This suggests that a structured agentic architecture augmented with domain-specific tools (e.g., the *Communication Expert*) can mitigate parameter disparity, providing a resource-efficient alternative for high-level video reasoning.

**Superiority Over Reasoning-Enhanced Models.** AD-MIR also surpasses reasoning-centric baselines like ReAd-R. While ReAd-R shows strong latent reasoning (51.5% relaxed), its strict accuracy collapses to 25.0%, highlighting the difficulty of aligning reasoning with rigid output constraints through standard optimization. Conversely, AD-MIR (Qwen2.5-VL-7B) improves over ReAd-R by +5.7 strict points while maintaining a comparable relaxed score (+0.6). This validates our training-free paradigm: instead of relying on potentially unstable fine-tuning, AD-MIR leverages structured tool usage to guide the frozen backbone, proving that architectural scaffolding offers a more robust mechanism for enforcing verifiable decision-making.

**Statistical Reliability and Human Reference.** Using question-level paired bootstrap over the official test-set predictions, AD-MIR(o1) remains ahead of DVD, with a

strict improvement of +1.8 points (95% CI: +0.28 to +3.31, $p \approx 0.020$) and a relaxed improvement of +9.5 points (95% CI: +7.94 to +11.04, $p < 10^{-16}$). Human annotators achieve 51.3% strict and 71.4% relaxed accuracy under the same scoring protocol, indicating that AdsQA remains far from saturated and that the remaining gap primarily lies in subtle cultural, rhetorical, and audience-inference questions; details are provided in Appendix B.1.

### 3.3. Ablation Study and Sensitivity Analysis

Using Qwen2.5-VL-7B, we ablate both hyperparameters and architectural modules in Figure 3; full per-dimension numbers are provided in Appendix C. Accuracy improves as the lexical weight $\beta$ increases and remains stable across a broad high-value range, confirming the importance of exact OCR and ASR cues that semantic retrieval can miss. Varying $T_{max}$ shows a similar pattern: multi-hop reasoning helps until $T_{max} = 8$, after which extra steps offer little benefit and may introduce noise. Component ablations further show that Hybrid Indexing preserves fine-grained text evidence, Subject Registry reduces distractor interference, Communication Expert contributes most to persuasion-centric dimensions, and Visual Anchor verification is critical for suppressing plausible but unsupported rationales.

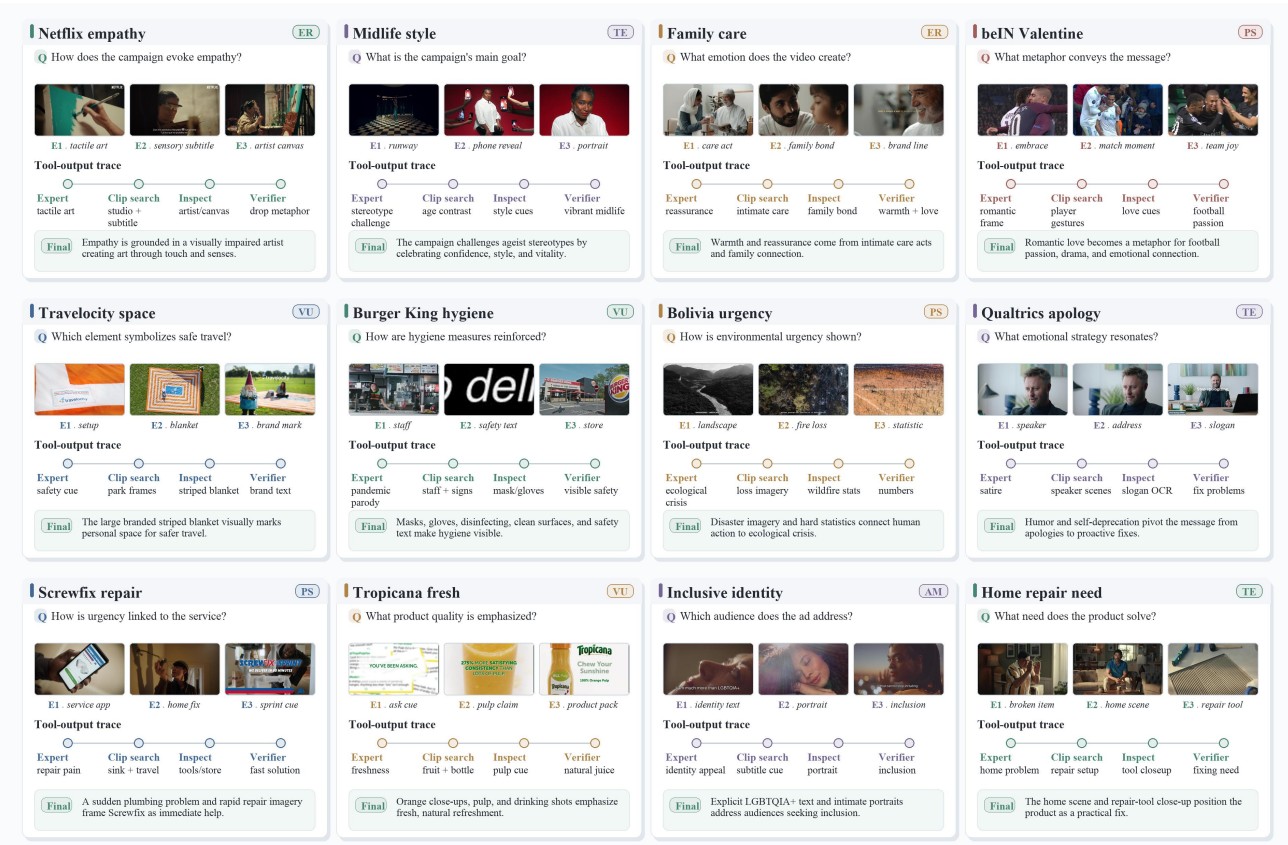

Figure 4. **Cross-video qualitative case gallery.** Each card shows a strictly correct AdsQA sample with inspected visual anchors, a compact trace summary, and the grounded final answer.

### 3.4. Efficiency and Qualitative Analysis

**Efficiency.** AD-MIR adds reasoning overhead relative to DVD because it performs explicit expert analysis and post-hoc grounding repair. On AdsQA, the full-run average latency is 93.1s per question for AD-MIR versus 45.5s for DVD, and the average token usage is 174.6k versus 55.9k over normal model/tool calls. This cost is concentrated in FRAME INSPECT and COMM. EXPERT, whereas the hybrid CLIP SEARCH itself is cheaper than DVD's dense top-$k$ retrieval (0.3s vs. 1.2s; 192 vs. 1.2k tokens). The full breakdown is shown in Figure 7.

**Qualitative Behavior.** Figure 4 shows representative full-evaluation cases where compact trace summaries connect hypotheses to retrieved clips, inspected anchors, verification feedback, and grounded final answers. Additional auditable traces appear in Appendix E.

## 4. Limitations

AD-MIR trades computation for stronger grounding. The expert-and-verifier loop roughly doubles latency compared with DVD and increases token use, which may be excessive for real-time moderation or large-scale indexing. In addition, advertising intent is culturally situated: slogans, humor, celebrity references, and public-service conventions can vary by region, so the Communication Expert may still need localization for markets underrepresented in current multimodal training data. Finally, AD-MIR depends on the quality of upstream captions, ASR, OCR, and the official LMM-judge protocol; when these perception layers miss non-English text or low-resolution brand cues, or when the judge misses a valid paraphrase, downstream measurement and reasoning can only partially recover.

## 5. Conclusion

AD-MIR demonstrates that advertising-intent reasoning benefits from being treated as evidence-grounded video understanding rather than direct answer generation. By combining structured memory, communication-expert analysis, targeted retrieval, frame inspection, and visual-anchor verification, the system turns implicit persuasive cues into auditable reasoning traces. The full AdsQA evaluation shows that this tool-grounded design improves strict and relaxed accuracy while making unsupported answers visible.

## Impact Statement

This work supports advertising audit and regulatory review by grounding implicit persuasive intent in video evidence. Because the same analysis could optimize manipulative targeting, deployment should emphasize transparency, consent, and compliance.

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

# A. Related Work

**Advertising Understanding** Computational advertisement understanding has long recognized that ads are deliberately engineered persuasive artifacts rather than objective records, making the key challenge to infer persuasive intent beyond literal content. Hussain et al. (2017) formalize this view by connecting observable cues (objects, scenes, edits) to symbolic references and persuasive strategies in large-scale image and video ad datasets. Classical marketing and persuasion theories further show that advertisements often stage attention, desire, credibility, social proof, and peripheral affective cues rather than simply displaying product attributes (Strong, 1925; Cialdini, 2001; Petty & Cacioppo, 1986). Building on this semiotic framing, our work focuses on modern information-dense advertising videos and emphasizes intent reasoning that leverages domain knowledge (marketing psychology, rhetoric) together with stricter evidence grounding to avoid plausible-but-unsupported narratives.

**Long-Form Video Understanding with Multimodal LLMs and Benchmarks** While the field of visual generation has witnessed rapid advancements in controllable autoregressive learning (Xu et al., 2025; Jin et al., 2025), precise scene text editing (Lan et al., 2025), and high-fidelity video synthesis applications (Wu et al., 2025; Zeng et al., 2025), the domain of *understanding* long-form narratives faces distinct challenges. Despite rapid progress in multimodal LLMs for video description and QA, long-form understanding remains bottlenecked by limited context, temporal fragmentation, and imperfect multimodal integration. Representative advances include unified frameworks like Video-LLaVA (Lin et al., 2024) and memory-augmented models such as MA-LMM (He et al., 2024) and PAM (Lin et al., 2025). However, recent benchmarks, including Video-MME, LVBench, ALLVB, MMBench-Video and MME-CoF (Fu et al., 2025; Wang et al., 2025; Tan et al., 2025; Fang et al., 2024; Guo et al., 2025), collectively reveal that current systems still degrade markedly on multi-hour videos and require explicit long-term memory. Motivated by these findings, our approach treats advertising videos as high-density long-form narratives, incorporating structured memory, temporal indexing, and iterative evidence collection tailored to intent-centric reasoning.

**Tool-Augmented Agents and Hallucination-Aware Multimodal Reasoning** A promising solution to long-context constraints is tool-augmented, agentic reasoning that iteratively gathers evidence and revises hypotheses. On the language side, Yao et al. (2022); Schick et al. (2023) show that LLMs can interleave reasoning with external tool calls and even self-supervise tool-use policies, while in the video domain Zhang et al. (2025); Fan et al. (2024) develop agentic frameworks such as DVD and VideoAgent for multi-granular retrieval and memory-augmented long-horizon understanding, and hallucination analyses including HallusionBench and object-centric evaluations like POPE and Logical Closed Loop by Guan et al. (2024); Li et al. (2023); Wu et al. (2024) demonstrate that LVLMs frequently produce confident yet weakly supported explanations. Building on this line of work, our method instantiates an advertising-specific tool-augmented agent with domain-adaptive inductive biases and hallucination-aware self-correction, explicitly enforcing evidence-first reasoning when mapping pixel-level anchors to persuasion-level intents.

# B. Implementation Details

In this section, we provide a detailed overview of the experimental setup, including the testing protocols for baseline models and the specific hyperparameter configurations for our proposed ADMIR agent.

## B.1. Baseline Testing Protocol

Following the benchmarking standards established in AdsQA, we evaluate baseline Multimodal Large Language Models (MLLMs) using a standardized zero-shot setting to ensure fair comparison.

**Frame Sampling Strategy.** For end-to-end MLLM baselines, we adopt a uniform sampling strategy. Given the high temporal redundancy in advertisement videos, we sample $N = 16$ frames uniformly distributed across the video duration. For models that support high-resolution inputs or dynamic resolution (e.g., o1), frames are resized while maintaining the aspect ratio, with the longest side not exceeding 1024 pixels.

**Prompting Strategy.** We utilize the standard prompts provided by the AdsQA benchmark. The prompt structure consists of a system instruction defining the task (Video QA), followed by the visual inputs (interleaved frames) and the specific question. For open-ended generation, we employ a greedy decoding strategy to ensure reproducibility.

**Human Reference and Statistical Testing.** We report the AdsQA human-reference scores under the same official strict and partial-credit relaxed metrics. Confidence intervals and $p$-values for method comparisons are computed from paired

question-level score differences over the official test set.

## B.2. AD-MIR Agent Configuration

The implementation details for the data preprocessing and agent hyperparameters are strictly aligned with the provided logic.

**Video Database Construction.** We preprocess the AdsQA video corpus to build a structured retrieval database:

- **Frame Extraction**: Videos are decoded at a frame rate of 1 FPS ('VIDEO_FPS=1').
- **Captioning and OCR**: We generate dense captions for video clips with a duration of 5 seconds ('CLIP_SECS=5'). OCR is performed by the same visual backbone through the fine-grained extraction prompt rather than by a separate OCR engine. In AD-MIR(o1), captioning/OCR uses `gpt-4o`; in AD-MIR(Qwen2.5-VL-7B), all captioning, OCR, and visual reasoning calls are replaced with `Qwen2.5-VL-7B`.
- **ASR**: Audio transcripts are extracted with Whisper ASR and aligned to timestamps before being inserted into the multimodal database.
- **Subject Registry**: A dynamic subject registry is maintained and merged across clips to track character identities and key objects.
- **Embeddings**: We utilize `BAAI/bge-m3` (dimension 1024) for local vector storage.

**Agent Hyperparameters.** We report two instantiations with identical control flow and retrieval hyperparameters. In AD-MIR(o1), the ReAct controller is orchestrated by `gpt-4o`, the specialized COMMUNICATION EXPERT uses `o1`, and the format-only answer refinement and visual-anchor repair use `gpt-4o-mini`. In AD-MIR(Qwen2.5-VL-7B), these LMM/VLM components are all replaced with `Qwen2.5-VL-7B`; the ASR model and embedding model remain Whisper and `BAAI/bge-m3`, respectively.

- **Maximum Iterations**: The ReAct loop is capped at $T_{max} = 8$ steps to prevent infinite loops, though most queries are resolved within 4–6 steps.
- **Global Browse**: The top $K = 40$ semantic captions ('GLOBAL_BROWSE_TOPK') and aligned ASR snippets are retrieved, chronologically ordered, and summarized during the pre-emptive global browse phase to construct the initial narrative context.
- **Clip Search**: A hybrid search mechanism (Vector + Keyword) retrieves the top $K = 5$ to 8 relevant clips for fine-grained inspection.
- **Expert Analysis**: The COMMUNICATION EXPERT processes up to 64 frames stitched into $2 \times 2$ grid images to analyze visual rhetoric and persuasion strategies.
- **Output Verification and Refinement**: The final output is checked against concrete visual anchors and compressed to strictly adhere to the token limit (under 30 words) while preserving critical entities and attributes.

**Retry Mechanism.** To handle API instability and potential hallucinations in tool calls, we implement an exponential backoff retry mechanism with a maximum of 8 retries for critical API failures.

## B.3. Algorithm Robustness Details

To prevent the ReAct agent from falling into infinite loops or overwriting high-confidence expert insights with low-level hallucinations, we implement three specific algorithmic safeguards within the inference controller.

**1. Temporal Stagnation Detection & Redirection.** In long-horizon reasoning, agents often become fixated on a specific time segment, repeatedly searching the same interval despite finding no new evidence. To mitigate this, we implement a *Temporal Stagnation Check*. Let $\mathcal{H}_{time} = \{(t_{start}^{(i)}, t_{end}^{(i)})\}$ be the history of time ranges queried by the agent. For a new query range $[t_s, t_e]$ with duration $d = t_e - t_s$, we calculate the overlap ratio with history:

$$Overlap(i) = \frac{\text{intersection}([t_s, t_e], [t_{start}^{(i)}, t_{end}^{(i)}])}{d} \tag{6}$$

If $Overlap(i) > 0.6$ for more than $N = 2$ distinct historical queries, the system triggers a **Forced Redirection Intervention**. The agent's query is overridden, and the focus is mechanically shifted to the video boundaries (First 15s or Last 15s) to break the cognitive deadlock.

**2. Evidence-Gated Source Hierarchy.** A common failure mode in hierarchical agents is "over-verification," where the

agent uses a sparse literal inspection to reject a high-level persuasive hypothesis that is supported by broader narrative evidence. The opposite failure mode is also dangerous: accepting an expert hypothesis as fact without visual support. We therefore use an evidence-gated hierarchy. The Communication Expert may propose abstract interpretations, but concrete claims (brand names, slogans, objects, actions, and timestamps) must be traceable to `clip_search`, `frame_inspect`, ASR, OCR, or explicit frames inside the expert grid. The controller is instructed to finish only after the answer contains sufficient anchors, and to re-query when an expert claim lacks concrete support. This prevents a low-level tool from falsely rejecting a persuasion label merely because the label is abstract, while still requiring visual-anchor verification for the factual evidence behind that label.

**3. Visual Anchor Repair.** For questions explicitly demanding visual grounding (e.g., "What specific object..."), the agent's draft answer is passed through the configured lightweight refinement module. If the answer lacks concrete visual nouns (anchors) found in the retrieval evidence, a repair loop is triggered to inject specific observable details (e.g., changing "a vehicle" to "a red Ferrari") before final output.

### B.4. Expert Model Input Strategy

To bridge the gap between token limits and the need for high temporal resolution in advertising analysis, we employ a **Spatio-Temporal Grid Projection** strategy for the `communication_expert_tool`.

**High-Density Sampling.** Unlike standard approaches that sample 8-16 frames, our expert module samples $N = 64$ frames uniformly distributed across the video duration (or the queried segment). This high density is crucial for capturing fleeting subliminal cues typical in commercials (e.g., split-second micro-expressions or flash cuts).

**2x2 Grid Stitching.** Directly feeding 64 images exceeds the file count limits of most VLM APIs. We therefore stitch consecutive frames into $2 \times 2$ grid images.

- **Batching**: The 64 frames are divided into 16 batches of 4 frames each.
- **Composition**: Each batch is stitched into a single high-resolution image ($H_{grid} = 2 \times H_{frame}, W_{grid} = 2 \times W_{frame}$).
- **Resolution**: The resulting grids are encoded with `detail: "high"` to preserve OCR-readability in sub-frames.

**Temporal Serialization.** The VLM receives a sequence of 16 grid images. To ensure correct temporal reasoning, the system prompt explicitly defines the reading order:

$$\text{Order: Top-Left} \rightarrow \text{Top-Right} \rightarrow \text{Bottom-Left} \rightarrow \text{Bottom-Right} \tag{7}$$

This projection allows the model to process 64 frames of visual information while only consuming the request overhead of 16 images, effectively quadrupling the temporal context window.

## C. Extended Quantitative Analysis

The relaxed component ablations in Figure 5 complement the strict ablations and sensitivity curves in Figure 6, showing that the same architectural components matter under both official scoring views.

Table 6: Relaxed component ablation on AdsQA. Full is the complete AD-MIR model. Drop is the absolute decrease from Full after removing one component.

| Task | Full | w/o Hybrid | Drop | w/o Subject | Drop | w/o Comm. | Drop | w/o Anchor | Drop |
|------|------|-----------|------|-------------|------|-----------|------|------------|------|
| VU | 52.5 | 48.0 | 4.5 | 48.7 | 3.8 | 49.7 | 2.8 | 48.3 | 4.2 |
| ER | 55.9 | 51.1 | 4.8 | 52.9 | 3.0 | 51.1 | 4.8 | 52.4 | 3.5 |
| TE | 53.0 | 48.0 | 5.0 | 49.1 | 3.9 | 49.8 | 3.2 | 48.4 | 4.6 |
| PS | 50.2 | 46.0 | 4.2 | 47.2 | 3.0 | 45.4 | 4.8 | 46.2 | 4.0 |
| AM | 50.1 | 45.9 | 4.2 | 45.7 | 4.4 | 46.6 | 3.5 | 46.1 | 4.0 |
| Avg. | 52.3 | 47.8 | 4.5 | 48.7 | 3.6 | 48.5 | 3.8 | 48.3 | 4.1 |

*Figure 5.* **Detailed relaxed component ablations.** The figure reports per-dimension relaxed accuracy after removing Hybrid Indexing, Subject Registry, Communication Expert, and Visual Anchor modules.

Table 1: Strict accuracy (%) under different $T_{\max}$ values. Best reports the peak over the six settings; Span reports the absolute best–worst change.

| Task | 2 | 4 | 6 | 8 | 10 | 12 | Best | Span |
|------|------|------|------|------|------|------|------|------|
| VU | 30.0 | 31.3 | 31.9 | 32.4 | 32.3 | 32.1 | 32.4 | 2.4 |
| ER | 32.1 | 33.3 | 34.3 | 34.8 | 34.7 | 34.5 | 34.8 | 2.7 |
| TE | 29.2 | 30.3 | 31.2 | 31.7 | 31.6 | 31.4 | 31.7 | 2.5 |
| PS | 27.4 | 28.7 | 29.5 | 30.1 | 29.9 | 29.7 | 30.1 | 2.7 |
| AM | 25.8 | 26.6 | 27.4 | 27.9 | 27.7 | 27.5 | 27.9 | 2.1 |
| Avg. | 28.9 | 30.0 | 30.9 | 31.4 | 31.2 | 31.0 | 31.4 | 2.5 |

Table 2: Relaxed accuracy (%) under different $T_{\max}$ values. Best reports the peak over the six settings; Span reports the absolute best–worst change.

| Task | 2 | 4 | 6 | 8 | 10 | 12 | Best | Span |
|------|------|------|------|------|------|------|------|------|
| VU | 51.5 | 53.1 | 54.2 | 54.9 | 54.7 | 54.4 | 54.9 | 3.4 |
| ER | 56.0 | 57.3 | 58.1 | 58.7 | 58.6 | 58.3 | 58.7 | 2.7 |
| TE | 52.6 | 54.1 | 54.9 | 55.4 | 55.3 | 55.0 | 55.4 | 2.8 |
| PS | 48.6 | 50.6 | 51.5 | 52.2 | 52.1 | 51.7 | 52.2 | 3.6 |
| AM | 48.9 | 50.3 | 51.2 | 52.1 | 51.9 | 51.5 | 52.1 | 3.2 |
| Avg. | 51.5 | 53.1 | 54.0 | 54.7 | 54.5 | 54.2 | 54.7 | 3.1 |

Table 3: Strict accuracy (%) under different lexical weights $\beta$. Best reports the peak over the six settings; Span reports the absolute best–worst change.

| Task | 0.0 | 0.2 | 0.4 | 0.6 | 0.8 | 1.0 | Best | Span |
|------|------|------|------|------|------|------|------|------|
| VU | 28.5 | 30.2 | 31.3 | 31.7 | 32.1 | 32.4 | 32.4 | 3.9 |
| ER | 31.2 | 32.7 | 33.9 | 34.2 | 34.5 | 34.8 | 34.8 | 3.6 |
| TE | 27.7 | 29.4 | 30.7 | 31.1 | 31.4 | 31.7 | 31.7 | 4.0 |
| PS | 27.0 | 28.6 | 29.6 | 29.8 | 29.9 | 30.1 | 30.1 | 3.1 |
| AM | 25.1 | 26.6 | 27.4 | 27.6 | 27.7 | 27.9 | 27.9 | 2.8 |
| Avg. | 27.9 | 29.5 | 30.6 | 30.9 | 31.1 | 31.4 | 31.4 | 3.5 |

Table 4: Relaxed accuracy (%) under different lexical weights $\beta$. Best reports the peak over the six settings; Span reports the absolute best–worst change.

| Task | 0.0 | 0.2 | 0.4 | 0.6 | 0.8 | 1.0 | Best | Span |
|------|------|------|------|------|------|------|------|------|
| VU | 50.3 | 52.0 | 53.4 | 53.8 | 54.3 | 54.9 | 54.9 | 4.6 |
| ER | 54.5 | 56.3 | 57.5 | 57.8 | 58.3 | 58.7 | 58.7 | 4.2 |
| TE | 51.4 | 53.1 | 54.3 | 54.6 | 55.1 | 55.4 | 55.4 | 4.0 |
| PS | 48.3 | 50.4 | 51.3 | 51.5 | 51.9 | 52.2 | 52.2 | 3.9 |
| AM | 48.0 | 49.9 | 50.9 | 51.3 | 51.7 | 52.1 | 52.1 | 4.1 |
| Avg. | 50.5 | 52.3 | 53.5 | 53.8 | 54.3 | 54.7 | 54.7 | 4.2 |

Table 5: Strict component ablation on AdsQA. Full is the complete AD-MIR model. Drop is the absolute decrease from Full after removing one component.

| Task | Full | w/o Hybrid | Drop | w/o Subject | Drop | w/o Comm. | Drop | w/o Anchor | Drop |
|------|------|------------|------|-------------|------|-----------|------|------------|------|
| VU | 31.8 | 30.3 | 1.5 | 30.6 | 1.2 | 31.0 | 0.8 | 30.3 | 1.5 |
| ER | 33.6 | 31.8 | 1.8 | 32.6 | 1.0 | 31.7 | 1.9 | 32.6 | 1.0 |
| TE | 31.3 | 29.2 | 2.1 | 29.8 | 1.5 | 30.2 | 1.1 | 29.6 | 1.7 |
| PS | 30.1 | 28.6 | 1.5 | 29.1 | 1.0 | 28.3 | 1.8 | 28.8 | 1.3 |
| AM | 28.4 | 27.1 | 1.3 | 26.8 | 1.6 | 27.3 | 1.1 | 27.0 | 1.4 |
| Avg. | 31.0 | 29.4 | 1.6 | 29.8 | 1.3 | 29.7 | 1.3 | 29.7 | 1.4 |

1

*Figure 6.* **Detailed strict ablations and sensitivity analyses.** The figure reports per-dimension strict accuracy under different reasoning budgets $T_{\max}$ and lexical weights $\beta$, together with strict component ablations.

## D. Efficiency Breakdown

Figure 7 decomposes the additional latency and token cost introduced by expert reasoning and grounding repair.

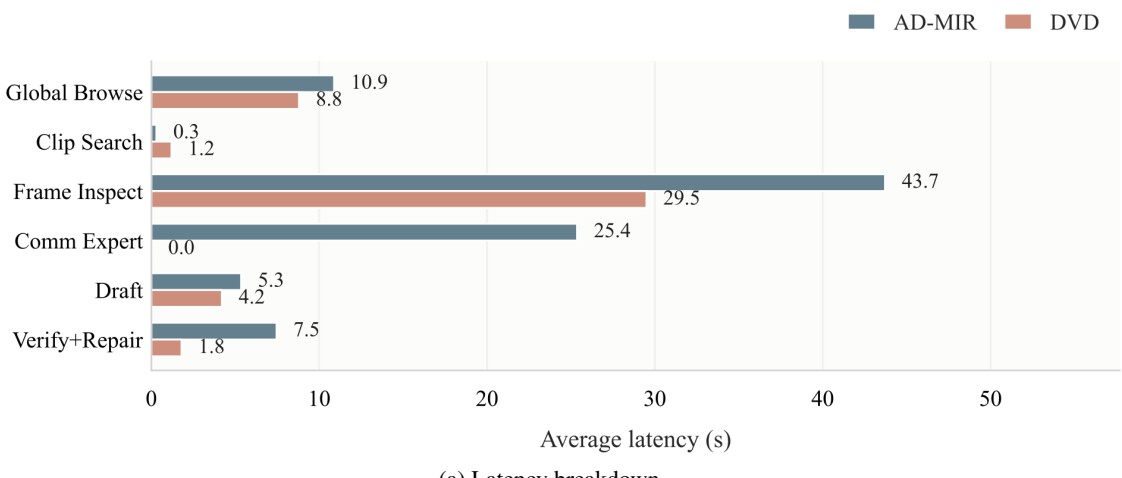

(a) Latency breakdown.

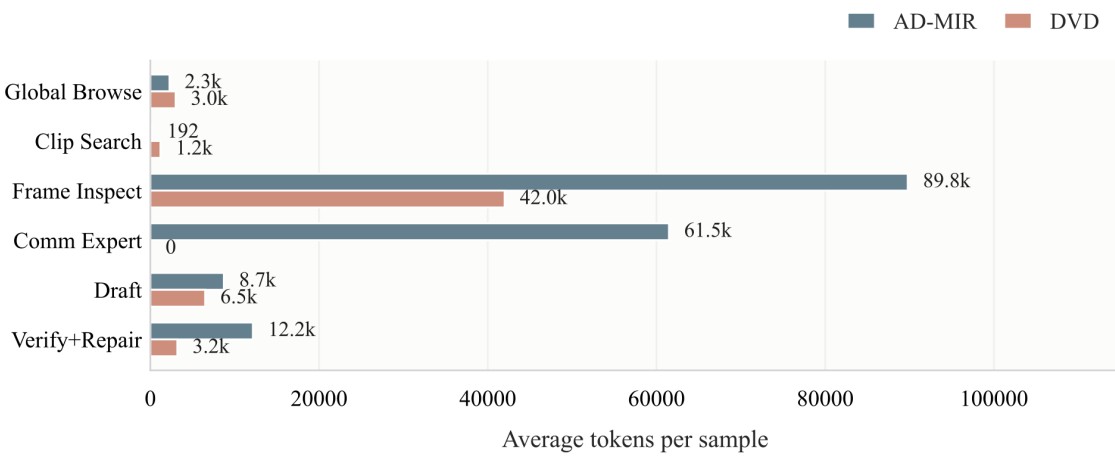

(b) Token breakdown.

*Figure 7.* **Efficiency comparison between AD-MIR and DVD on AdsQA.** AD-MIR is more expensive overall because it adds expert reasoning and grounding repair, while its hybrid clip search is cheaper than DVD's dense retrieval.

# E. Qualitative Case Trajectories

Figure 8 gives the compact case-level overview, while Figures 9 and 10 expand two cases into auditable tool-output traces.

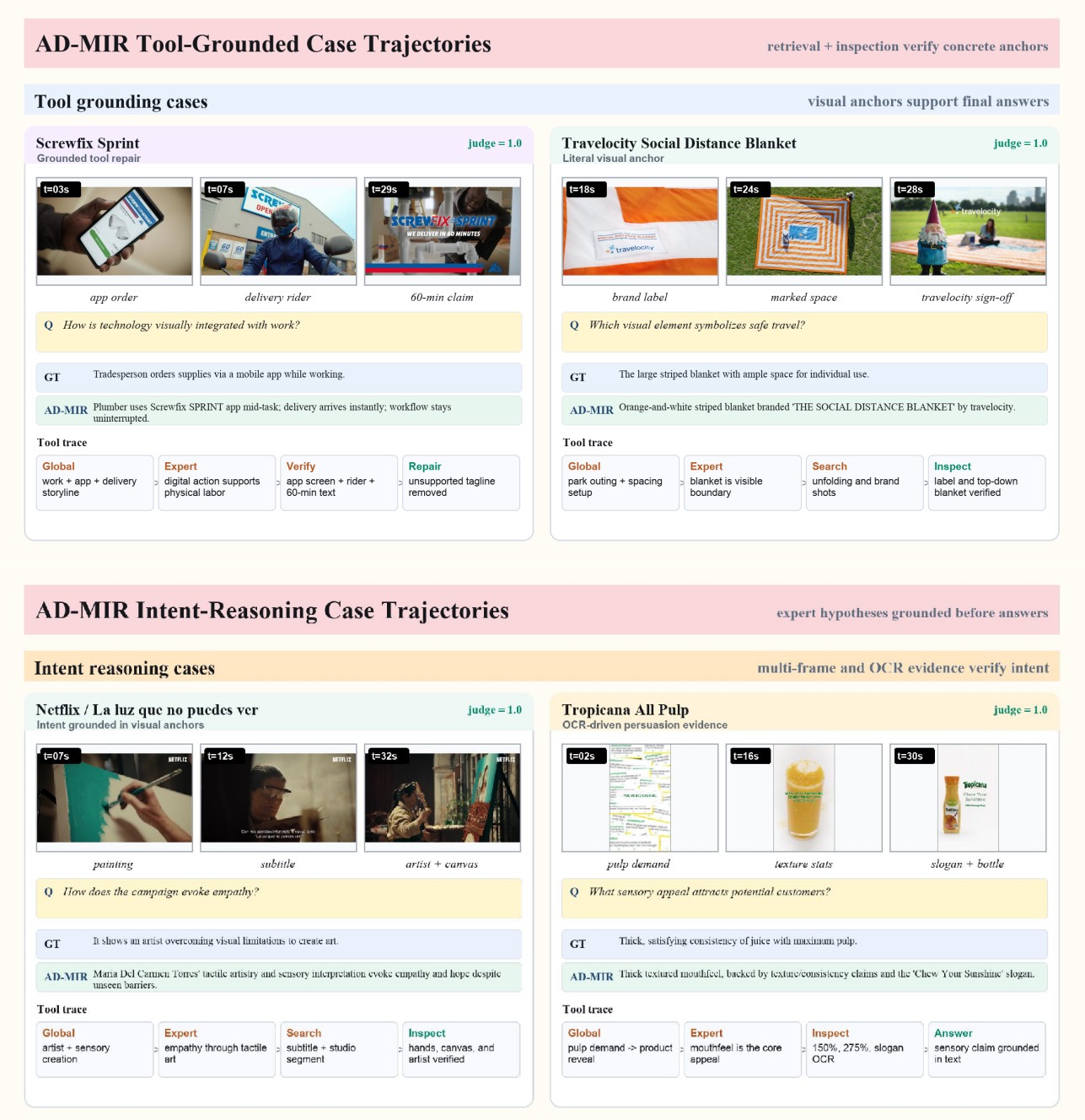

*Figure 8.* **Qualitative examples of AD-MIR reasoning on AdsQA.** We select four cases from the full evaluation for which AD-MIR receives a strict official-judge score of 1.0. Each card shows three retrieved or inspected video frames, the question, the official ground-truth answer, AD-MIR's final answer, and the tool-level evidence trace used to support that answer. The cases highlight answer repair after verification, literal visual-anchor grounding, intent reasoning over visual evidence, and OCR-grounded persuasion evidence.

**Auditable Tool-Output Traces.** Figures 9 and 10 expand two representative cases into step-by-step audit traces. Each row reports the tool request, a concise summary of the logged tool output, the evidence IDs linked to retrieved or inspected frames, and the verifier's effect on the final answer. These figures expose the externally logged reasoning artifacts used by AD-MIR, rather than hidden chain-of-thought.

## F. Core Prompts of AD-MIR

This appendix collects the prompt templates that instantiate AD-MIR's perception, tool-use, repair, and evaluation stages. Colored labels mark each template's role, target behavior, and guardrails, while the shaded blocks preserve the verbatim instructions used in our experiments.

### F.1. Perception & Context Construction Prompts

The following prompts drive the offline perception layer, transforming raw video signals into structured semantic indices through narrative reconstruction and dynamic inspection.

---

**Global Browse: Text-Based Forensic Reconstruction**

ROLE   Media Forensics Expert

GOAL   Reconstruct the narrative and ground truth from multimodal logs (Visual Captions + Audio) without direct pixel access.

KEY MECHANISMS

- **A/V Cross-Referencing**: Combining visual descriptions with audio transcripts to infer specific entities (e.g., "Product shown" + "Audio mentions Pepsi" → "Pepsi").
- **Text Hunting**: Specifically scanning logs for [POTENTIAL_TEXT] tags to extract explicit on-screen text.

SYSTEM PROMPT

```
You are a Media Forensics Expert specialized in solving AdsQA cases.
**YOUR CONSTRAINT**: You cannot see the video. You only have access to:
1. **Visual Logs**: Text descriptions of what happens.
2. **Audio Transcript**: What was said.

**STRATEGY FOR TEXT-ONLY ANALYSIS**:
1. **Cross-Referencing**: If Visual Log says "[SCENE]: A product is shown" and Audio says "Try the new Pepsi", you
     must infer the product is "Pepsi".
2. **Text Hunting**: Look specifically for the `[POTENTIAL_TEXT]` tags in the logs.
3. **Implicit Clues**: Identify brands/objects from specific descriptions (e.g., red car + horse logo $\to$ Ferrari
     ).

**OUTPUT FORMAT (JSON)**:
{
  "narrative_reconstruction": "Story flow (Hook $\to$ Middle $\to$ End).",
  "inferred_objects": ["List of objects implied by combining visual + audio"],
  "explicit_text_found": ["Text explicitly quoted in visual logs"],
  "audio_visual_mismatch": "Contradictions between seen and heard?",
  "final_answer": "Direct answer to the USER QUERY."
}
```

---

**Frame Inspect: Dynamic Mode Switching**

FUNCTION   'frame_inspect_tool'

STRATEGY   The tool dynamically selects an analysis mode based on the query type (Factual vs. Abstract).

MODE A: LITERAL INSPECTION (OCR & FACTUAL)

```
[MODE: LITERAL INSPECTION & OCR]
1. **VISUAL LOG**: Create a chronological log of events.
2. **TEXT**: Transcribe any visible text/logos verbatim.
3. **DETAILS**: List objects and specific physical interactions.
```

MODE B: SEMANTIC ANALYSIS (NARRATIVE & SYMBOLISM)

```
[MODE: SEMANTIC & NARRATIVE]
1. **STORY**: Describe the setup -> action -> outcome.
2. **TWIST**: Note if the environment changes artificially (e.g., reveal of a set).
3. **MEANING**: Identify key symbols and subject relationships.
```

---

**Clip Search: Semantic Query Rewrite**

**FUNCTION**  `clip_search_tool`

**GOAL**  Generalize specific user queries into broader visual concepts for better retrieval recall.

**REWRITE PROMPT**

```
Rewrite '{original_query}' into 3 simple, comma-separated keywords/phrases for video retrieval.
Rules: Keep it generic. Do NOT invent specific visual details (e.g. do NOT change 'luxury' to 'gold').
Input: 'sad man' -> Output: crying person, unhappy face, depressed mood
```

## F.2. Structured Agent Reasoning Prompts

These prompts control the online inference engine, managing the interaction between the ReAct controller and domain-specific experts.

---

**Communication Expert: Visual Semiotics Analysis**

**INPUT**  2x2 Grid Images (High-Density Temporal Sampling)

**CORE ANALYSIS PROTOCOLS**

- **2x2 Grid Reading**: Explicit instruction on reading order (TL → TR → BL → BR).
- **Grounding**: Strict prohibition against hallucinating objects not present in the grid.

**SYSTEM PROMPT**

```
You are an Elite Advertising Forensics Expert & Visual Semiotics Analyst.
**YOUR MISSION**: Decode the provided video content to uncover the narrative structure, character relationships,
    and persuasive strategy.

**INPUT FORMAT (CRITICAL)**:
- The visual input consists of **2x2 Grid Images**.
- Each image contains **4 chronological video frames**.
- **Reading Order**: Analyze each grid from **Top-Left -> Top-Right -> Bottom-Left -> Bottom-Right**.

**CORE ANALYSIS PROTOCOLS:**
1.  **OCR & BRAND TRUTH (HIGHEST PRIORITY)**: Any text on screen is fact. Identify Logos.
2.  **UNIVERSAL CHARACTER DYNAMICS**: Analyze Transactional, Conflict, or Affectionate interactions.
3.  **NARRATIVE ARC**: Hook -> Problem -> Product -> CTA.
4.  **EVIDENCE SEPARATION**: Separate directly observed evidence from inferred persuasion strategy.
5.  **GROUNDING**: Do not invent objects not present in the grids.
```

---

**ReAct Controller: System Instructions**

**ROLE**  Advanced Video Analysis Agent

**PHILOSOPHY**  Evidence-gated expert reasoning

**SYSTEM PROMPT**

```
You are an advanced Video Analysis Agent. Your goal is to answer the user's question precisely using the provided
    tools.

**CORE PHILOSOPHY: EVIDENCE-GATED EXPERT REASONING**
1.  **EXPERT AS HYPOTHESIS GENERATOR**: Use `communication_expert_tool` to propose high-level narrative and
    persuasion hypotheses.
2.  **VERIFY CONCRETE ANCHORS**: Brand names, visible text, objects, actions, and timestamps must be supported by `
    clip_search_tool`, `frame_inspect_tool`, ASR/OCR, or explicit frames in the expert grid.
3.  **DO NOT OVER-VERIFY ABSTRACT LABELS**: A sparse literal frame check should not reject a supported abstract
    strategy merely because the word itself is not visible.
4.  **FINISH ONLY WITH EVIDENCE**: Call finish only when the final answer is supported by sufficient concrete
    anchors.

**OUTPUT PROCESS**:
1.  **THOUGHT**: First, write a brief thought analysis in plain text explaining your reasoning.
2.  **ACTION**: Then, call the appropriate tool using the native function calling capability.
```

## Output Verification & Refinement

**STAGE** Post-Processing

**TASK** Compress output to $< 30$ tokens and repair visual grounding errors.

**REFINEMENT PROMPT**

```
You are compressing an answer for a visual QA benchmark.
Rewrite the answer to be <= 25 words, but DO NOT lose any core information.

[HARD CONSTRAINTS]
- Preserve ALL essential facts: names/entities, key attributes (color, number, time), and negation.
- DO NOT add new facts or hallucinations.
- If original is entity identification, output a compact noun phrase.

[EDITING RULES]
1) Remove meta phrases: "the video shows", "the answer is".
2) Keep specificity: numbers, brand names, proper nouns.
```

**VISUAL GROUNDING REPAIR PROMPT**

```
Rewrite the answer so it names ONE specific, observable scene/object/action from the evidence.
Hard constraints:
- Mention a concrete visual element (what is shown), not abstract traits.
- Use at least 1 anchor word if possible.
- Under 30 tokens.
```

### F.3. Data Preprocessing Prompts

The following prompts drive the offline perception layer, transforming raw video signals into structured semantic indices through narrative reconstruction, entity tracking, and fine-grained text extraction.

## Visual Captioning & Subject Registry Extraction

**INPUT** Consecutive Video Frames (Clip Level)

**CORE ANALYSIS PROTOCOLS**

- **Structured Output**: Strict JSON format enforcement for database indexing.
- **Subject Profiling**: Extract detailed appearance and identity for the context-anchored registry.

**SYSTEM PROMPT**

```
Here are consecutive frames from a video clip. Please visually analyze the video clip and output JSON in the
    template below.

Output template:
\{
  "clip\_start\_time": CLIP\_START\_TIME,
  "clip\_end\_time": CLIP\_END\_TIME,
  "subject\_registry": \{
    "<subject\_i>": \{
      "name": "<fill with short identity if name is unknown, e.g. 'man in red'>",
      "appearance": "<list of visual appearance descriptions>",
      "identity": "<list of inferred identity descriptions>",
      "first\_seen": "<timestamp>"
    \},
    ...
  \},
  "clip\_description": "<smooth and detailed visual narration of the video clip>"
\}
```

## Subject Registry Merging

**INPUT** List of Partial Subject Registries (from multiple clips)

**CORE ANALYSIS PROTOCOLS**

- **De-duplication**: Merge subjects referring to the same visual entity across time.

- **Union**: Combine attribute fields while preserving the earliest timestamp.

**SYSTEM PROMPT**

```
You are given several partial 'new\_subject\_registry' JSON objects extracted from different clips of the *same*
    video.

Task:
1. Merge these partial registries into one coherent 'subject\_registry'.
2. Preserve all unique subjects.
3. If two subjects visually refer to the same person/object, merge them (keep earliest 'first\_seen' time and union
    all fields).

Input (list of JSON objects):
REGISTRIES\_PLACEHOLDER

Return *only* the merged 'subject\_registry' JSON object.
```

## Fine-Grained OCR Extraction

**INPUT** Single Video Frame (High Resolution)

**CORE ANALYSIS PROTOCOLS**

- **Exhaustive Extraction**: Capture all visible text including logos, brand names, and background signs.
- **Format**: One item per line, strictly no visual descriptions.

**SYSTEM PROMPT**

```
Extract ALL visible text from this image.
Include: titles, labels, captions, signs, logos, brand names, slogans, any written content.
Return ONLY the extracted text, one item per line.
If no text is visible, return "NO\_TEXT".
Do not describe the image, only extract text.
```

## F.4. Evaluation Prompts

The prompt utilized by the LLM-as-a-Judge mechanism to benchmark performance against ground truth annotations.

## AdsQA Evaluation Judge

**ROLE** Advertising Expert Judge
**GOAL** Evaluate semantic alignment between prediction and golden answer.
**JUDGE PROMPT**

You are an advertising expert specializing in evaluating whether a respondent's answer after watching a video
    matches the golden answer. We will provide the video's Meta-Information, Question, Golden Answer, and the
    Response to be judged below.

###The meta-information includes the advertisement video's theme, creative points, and a brief content description,
    which can be regarded as ground-truth information, as follows::
{meta_info}

###Question:
{question}

###Golden Answer:
{golden_answer}

###Rule:
1. If the response to be judged contains ALL key information of the golden answer or expresses the same meaning
    using other sentences or synonyms, it is considered a match with the golden answer, and the output is 1.
2. If the response to be judged does NOT contain the key information from the golden answer, it is considered a
    mismatch, and the output is 0.
3. The response to be judged should NOT contain any content that is contradictory, conflicting, or unreasonable
    when inferred from the meta-information. If such content exist, it is considered a mismatch, and the output
    is 0.
4. If the response to be judged contains the MOST of key information of the golden answer and, do NOT contain any
    information that is contradictory, conflicting, or unreasonable when inferred from the meta-information, it
    is considered a partial match, and the output is 0.5.

###Response to be judged:
{response}

###Instructions:
Follow the format below and do not give any extra outputs:
Answer: 0 (if the response does not match)
Answer: 0.5 (if the response partially match)
Answer: 1 (if the response matches)

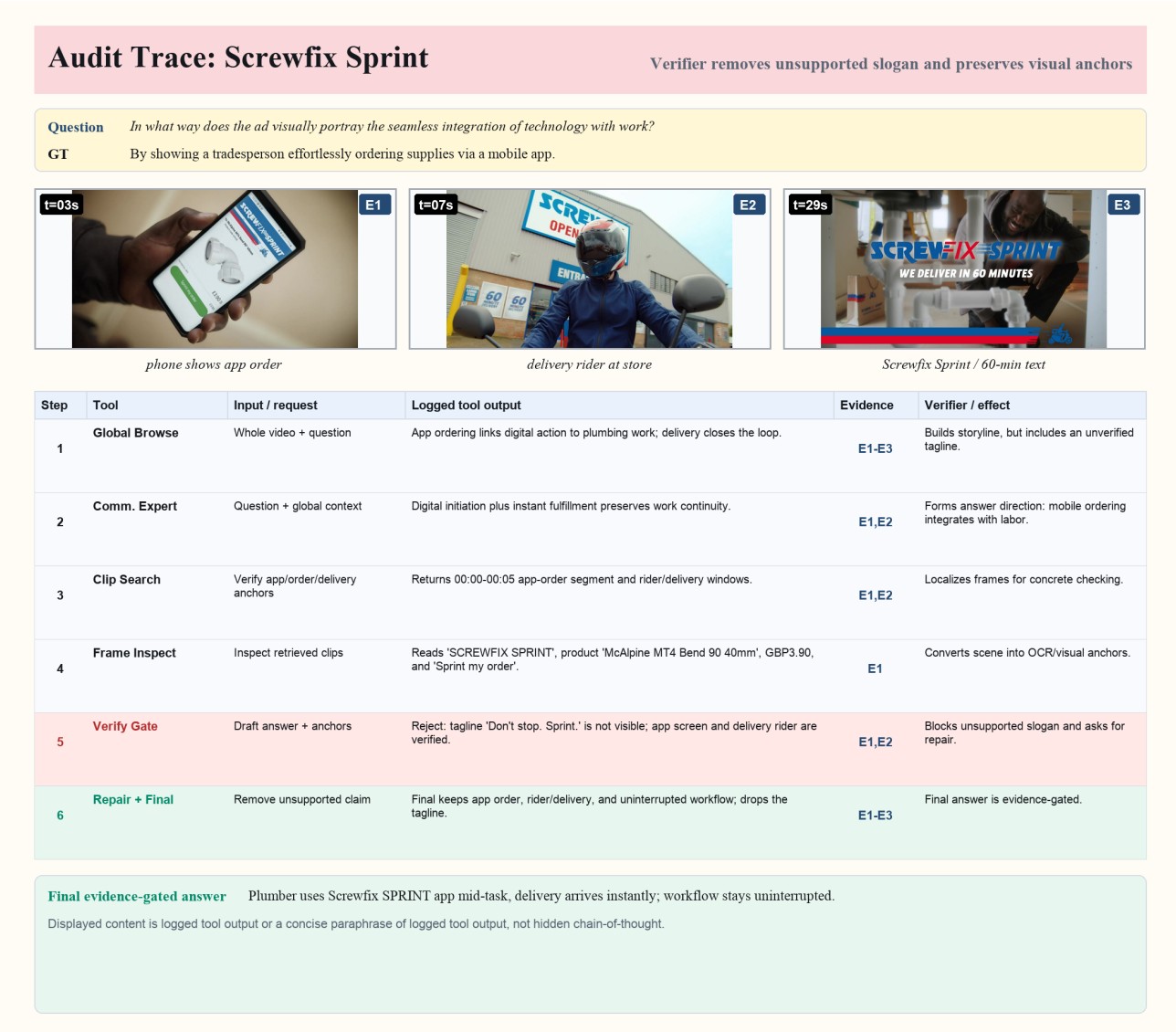

*Figure 9.* **Auditable trace for verifier-guided repair.** In the Screwfix case, the initial narrative and expert hypothesis correctly identify mobile app ordering and delivery as the visual mechanism, but the verifier rejects an unsupported slogan that is not visible in the inspected frames. The final answer removes the unsupported claim and keeps only the visually grounded app-order and delivery evidence.

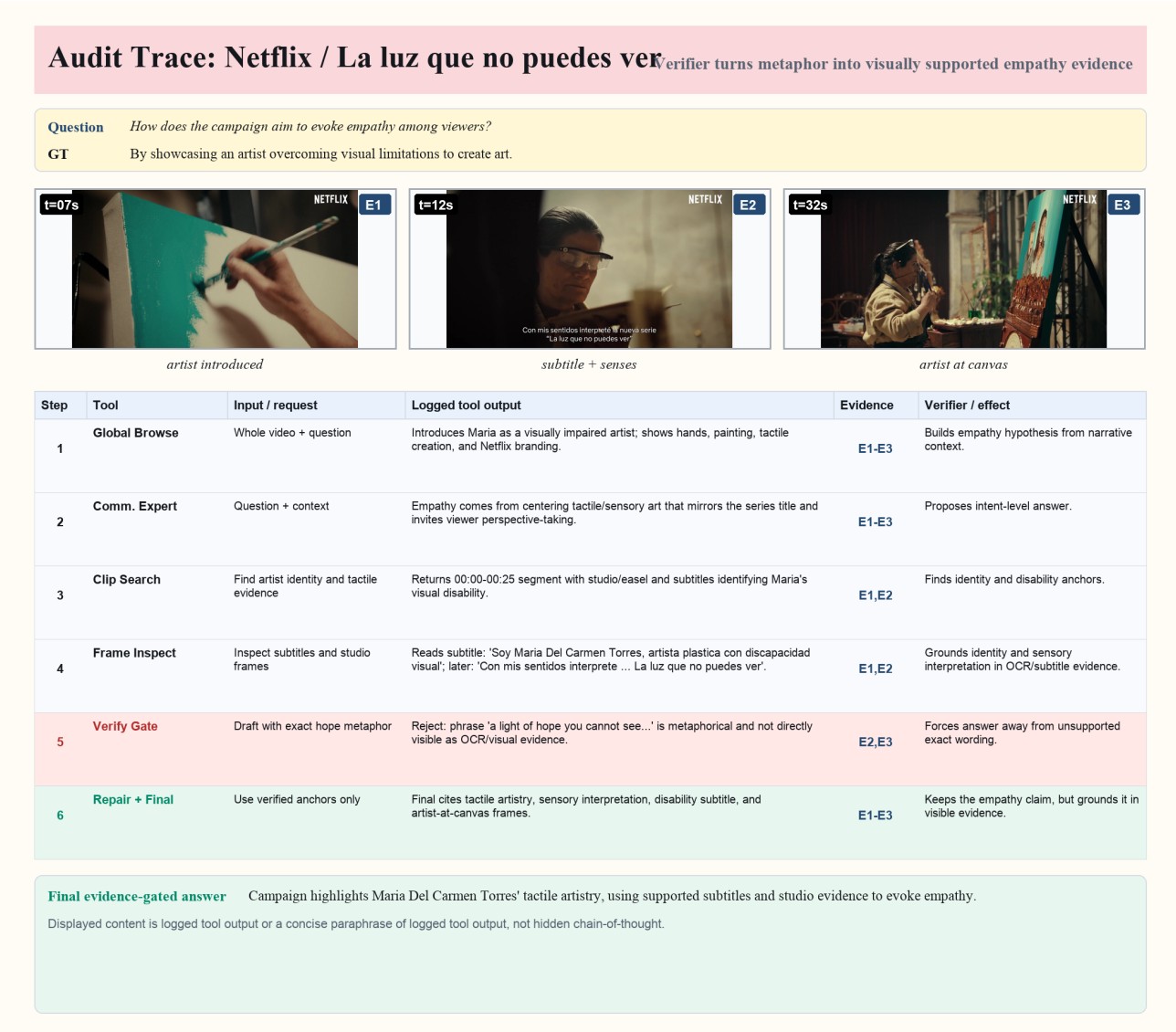

Figure 10. **Auditable trace for grounded intent reasoning.** In the Netflix case, AD-MIR first proposes an empathy interpretation from the campaign narrative. The verifier rejects exact metaphorical wording that is not directly supported by OCR or visual anchors, causing the final answer to rely on the verified artist identity, visual-disability subtitle, tactile art-making, and artist-at-canvas evidence.

