# OpenReview forum: "AD-MIR: Bridging the Gap from Perception to Persuasion in Advertising Video Understanding via Structured Reasoning"
_ICML.cc/2026/Conference — ICML 2026 regular_

### Official Review · Reviewer_29N3 · 2026-03-09

**Soundness:** 3
**Presentation:** 2
**Significance:** 2
**Originality:** 2
**Overall Recommendation:** 4
**Confidence:** 3

**Summary:**

This paper addresses the core pain point that existing multimodal models cannot bridge the gap between pixel-level perception and high-level marketing logic cognition in advertising video understanding, and proposes the AD-MIR two-stage multimodal reasoning framework.

**Compliance With Llm Reviewing Policy:**

Affirmed.

**Final Justification:**

Some of my concerns have been addressed. Having also reviewed the comments from other reviewers, I improve my score.

**Key Questions For Authors:**

Same as above weaknesses.

**Limitations:**

yes

**Strengths And Weaknesses:**

Strengths：


Weaknesses：
* The originality of core innovations is insufficient. The overall architecture is entirely based on ReAct method. And custom adaptations for the advertising field are only made in the toolset and pre-memory construction, without proposing core theoretical or architectural innovations. Essentially, it leans towards domain engineering optimization.
* Generalization verification is completely lacking.
* The core Communication Expert module relies entirely on frozen large models (o1/GPT-4o) and prompt engineering to achieve the decoding of marketing logic. The core ability to decode persuasion strategies is highly dependent on the base model itself, rather than the innovative contributions of the framework.
* Efficiency evaluation is completely lacking, and practical applicability is insufficient. The paper only focuses on accuracy metrics and does not conduct any evaluation on the inference time, token consumption, or computational resource overhead of AD-MIR. However, one of the core bottlenecks of Agent-related work is the efficiency issue in multi-round tool calls. The lack of efficiency evaluation greatly diminishes the industrial application value of the method.
* There is a contradiction in the design of the core mechanism. The designed Anti-Verification Protocol mandates unconditional trust in the output of the Communication Expert, abandons visual verification of the core reasoning process, and only performs shallow verification at the final answer generation stage, which is fundamentally contradictory to the core selling point claimed in the paper.

---

> ### Author Rebuttal · Authors · 2026-03-31
>
> Thanks for your thoughtful response and the time you have invested in reviewing our paper. After thoroughly reading the comments, we provide the following point-by-point responses：
> > W1: The originality of core innovations is insufficient.
>
> We clarify that AD-MIR solves core LMM limitations via fundamental architectural innovations, extending far beyond domain engineering. As Reviewer M6m8 recognized, it is a "novel agent framework":
>
> * **Macro-to-Micro Verification:** Unlike standard ReAct's hallucination-prone flat loop, we structurally decouple abstract hypothesis generation from pixel-level evidence gathering. Reviewer M6m8 noted this "hierarchical toolchain" mitigates hallucinations and is "the most difficult and appealing part."
> * **Adapting to Non-Linear Narratives:** What you termed "pre-memory construction" is actually our Hybrid Spatio-Temporal Indexing Architecture. It structurally reorganizes fragmented ad montages into a dual-searchable space. Reviewer zEkg highlighted this "hybrid semantic-lexical indexing is reasonable for ads" and praised the pipeline's modularity.
>
> > W2: Generalization verification is completely lacking.
>
> We clarify that AD-MIR's generalization was intrinsically verified across three dimensions, which we will consolidate into a dedicated "Generalization Analysis" subsection in the revision:
>
> * **Cross-Model:** The framework is model-agnostic, yielding consistent reasoning improvements across diverse families, from proprietary (o1: +5.0% strict) to open-source backbones (Qwen2.5-VL-7B: +4.5% strict).
> * **Cross-Task:** It elevates performance across all five diverse cognitive tasks, seamlessly handling both concrete factual retrieval (Visual Concept Understanding) and highly abstract intent deduction (Persuasion Strategy Mining).
> * **Cross-Domain:** The cognitive priors injected into the Communication Expert successfully generalized across 9 distinct commercial and public service domains (e.g., Automotive, Food, Health), proving robustness to highly varied visual distributions and narrative styles.
>
> > W3: The core ability to decode persuasion strategies is highly dependent on the base model itself, rather than the innovative contributions of the framework.
>
> We respectfully disagree. Empirical results and reviewers confirm our architecture drives performance, bridging the proprietary/open-source gap (Reviewer ApHm):
> * **Base Models Fail Without AD-MIR:** Zero-shot o1 achieves only 33.1% strict accuracy; AD-MIR improves this by +5.0%, activating dormant knowledge.
> * **Architectural Scaffolding:** Enforcing Two-Step Reasoning and Macro-to-Micro Verification provides strict grounding. Reviewers M6m8 and zEkg validated that this structure "creatively utilizes expert knowledge" and effectively grounds abstract concepts in visual evidence.
> * **Model-Agnostic Gains:** We achieved parallel strict accuracy gains (+4.5%) with the open-source Qwen2.5-VL-7B. Reviewer zEkg independently verified this, proving our framework's success relies on architecture, not just proprietary capacities.
>
> > W4: Efficiency evaluation is completely lacking, and practical applicability is insufficient.
>
> We provide detailed AD-MIR vs. DVD comparisons at `https://anonymous.4open.science/r/Anoymous_images-47FB/` (please see `latency_breakdown.png` and `token_breakdown.png`).
>
> While AD-MIR increases total latency (93.1s vs 45.5s) and tokens (174.6k vs 55.9k), this "computation-for-cognition" trade-off is highly justified:
> * **Cognitive Overhead:** The extra cost stems entirely from deep reasoning (`Comm Expert`, `Frame Inspect`, `Verify+Repair`), explicitly driving our massive accuracy leaps (+9.5% relaxed).
> * **Efficient Retrieval:** Conversely, our domain-specific `clip_search` is significantly faster (0.3s vs 1.2s) and cheaper (192 vs 1.2k tokens) than the DVD baseline, proving our infrastructure is fundamentally optimized rather than just stacking compute.
>
> We will add this practical trade-off analysis to the Appendix.
>
> > W5: There is a contradiction in the design of the core mechanism.
>
> We clarify that these protocols are not contradictory but form a **hierarchical verification design** to prevent category errors by strictly decoupling two cognitive dimensions (praised by Reviewers M6m8, zEkg, and ApHm):
>
> * **Anti-Verification Protocol (Abstract/Strategic):** Prevents low-level literal tools from falsely rejecting valid expert-level semantic deductions (e.g., metaphors, "fear appeal").
> * **Visual Grounding Verification (Concrete/Factual):** Demands strict pixel-level proof for concrete entities (e.g., "red Ferrari", OCR text), preventing the LLM from hallucinating claims that do not exist in retrieved frames.
>
> Together, they protect abstract strategic insights while enforcing rigorous evidence for factual claims. We will explicitly add this hierarchical breakdown to the revision.

---

> > ### Author Rebuttal · Reviewer_29N3 · 2026-04-04
> >
> > Thank you for your rebuttal. Some of my concerns have been addressed. Having also reviewed the comments from other reviewers, I will improve my score.

---

### Official Review · Reviewer_ApHm · 2026-03-10

**Soundness:** 3
**Presentation:** 2
**Significance:** 3
**Originality:** 3
**Overall Recommendation:** 4
**Confidence:** 3

**Summary:**

The authors present Ad-MIR, a framework focused on ad video understanding with a focus on capturing the higher-level cognitive intents of the advertisement. The framework shows SOTA performance on AdsQA dataset. Using ablation test the authors show the relevance of the different components of the framework for different aspects of higher level video understanding.

The framework involves four components – A structured data base, a react controller, information retrieval and reasoning tools and a verification module

 - Structured data base consists of an offline process of using visual descriptions (VLM) and audio transcripts (Whisper model) to get semantic embeddings of clips(video is divided into clips). Additionally, a subject registry (descriptions of different actors in the video extracted using GPT-4o) is combined with a query dependent anchor that helps to extract relevant actors and suppress background noise from the subject registry.

- The prompt guided ReAct controller is designed to use a Macro-to-Micro evidence zooming strategy to prevent disorientation and triggers a forced trajectory backtrack when visual grounding verification fails. The goal of this controller is to mimic cognitive processes of marketing experts.

- The information retrieval and reasoning tools include a communication expert that majorly contributes to connect visual data with persuasive intent, a global browse tool which uses the structured database to extract a structured summary of the video’s macro genre and important actors, the clip search tool that uses the modified query from the communication expert to identify candidate clips and create an event block and, a frame inspect tool that has two modes: literal, to extract factual data like object count and semantic mode, to interpret stylistic elements like lighting

 - Finally, the verification module evaluates the visual grounding in candidate answers (trajectory backtrack is initiated by the ReAct controller if this fails)

**Compliance With Llm Reviewing Policy:**

Affirmed.

**Final Justification:**

The authors have provided a good rebuttal. The quality of manuscript can be greatly improved if relevant cognitive literature and better description of key components like "Communication expert" are added to it. I maintain my original score.

**Key Questions For Authors:**

- The AdsQA dataset also contains human performance, in addition to comparing the models with other video understanding models maybe it would be interesting to compare the performance with human data as well ?

- You use DVD which is specifically developed for long videos (3hrs) as a baseline and also take inspiration from some of its methodology (breaking a video into clips to create a structured database). However, advertising videos are way shorter, do you think it is an overkill to use DVD inspired methods for ad videos ?

- Some sentences are not clear – “Conversely, AD-MIR (Qwen2.5-VL-7B) achieves a favorable balance (+5.7% strict, +0.6% relaxed) without parameter updates” – what is favorable ? and is it percentage or percentage points?

**Limitations:**

Limitations of the study in terms of applicability to real-world ad videos must be discussed.

**Strengths And Weaknesses:**

Strengths:

- The authors take inspiration from previous work like ReAct and DVD to develop a framework that achieves SOTA in video understanding for Advertisement videos (AdsQA dataset).

- The framework shows potential to bridge the gap between proprietary and open source models for ad video understanding

- Ablation test is done to show the relevance of the different modules and some hyperparameters which can be useful for future work extending this framework for different use cases. (Quantifying the results can make the analysis even stronger)

- Work in the direction of intent understanding of advertisement videos has a potential positive impact and can help with inspection and accountability.

Weaknesses:

- Communication expert which has been shown to contribute to persuasion metric in the ablation test must be explained and motivated in detail. Figure 2 contains additional details like “Step1:Perception,Step 2: Marketing strategy” which is not explained in the main text. How is the domain knowledge or marketing psychology used in this module ?

- The SOTA performance in PS is associated with using explicit marketing priors and the ReAct controller is said to emulate the cognitive process of marketing experts, however, the study lacks literature review or explanation of these concepts. For example, what marketing priors were used, or what type of cognitive processes of marketing experts were considered for replication in the controller.

- Use of some terms like “global browse”, “clip search” overlap with previous work (DVD) but have a different meaning. Either the differences/similarities (with reference to previous work) must be explicitly mentioned, or the tools must be referred by different names to avoid confusion for the readers.

---

> ### Author Rebuttal · Authors · 2026-03-31
>
> Thanks for your thoughtful response and the time you have invested in reviewing our paper. After thoroughly reading the comments, we provide the following point-by-point responses：
>
> > W1 & W2: Details on Figure 2 ("Step 1/2") and the specific marketing priors/cognitive processes.
>
> We will update Sec 2.3 to detail the Expert module and ground these concepts:
>
> **1. Emulating Experts:** We replicate top-down deductive reasoning via **Two-Step Reasoning** (Step 1: log literal visual facts; Step 2: deduce abstract strategies strictly constrained to those facts) and **Macro-to-Micro Zooming** (form a global hypothesis first, then direct `frame_inspect` to find pixel-level evidence).
>
> **2. Marketing Priors:** We inject training-free priors (App C.2) based on classical theory: **Narrative Arc** (AIDA Model), **Persuasive Framing** (Cialdini’s Principles), and **Character Dynamics** (Consumer Behavior).
> > W3: Either the differences/similarities (with reference to previous work) must be explicitly mentioned, or the tools must be referred by different names to avoid confusion for the readers.
>
> We agree that overlapping tool names with DVD causes confusion. Following your suggestion, we will retain the current names but add a dedicated paragraph in the revised Method section explicitly detailing how our algorithms are specifically redesigned for advertising compared to DVD's baseline:
>
> | Tool Name | DVD (Baseline) | AD-MIR (Our Implementation) |
> | :--- | :--- | :--- |
> | **global_browse** | Standard topic-level summary. | Synthesizes ASR and visual captions into a structural prior to prevent hallucinations. |
> | **clip_search** | Top-K dense semantic search. | Combines lexical/dense search with temporal event fusion to handle fragmented clips. |
> | **frame_inspect** | Standard VLM QA on a single frame. | Dual-mode inspection dynamically switching between literal OCR (factual) and semantic sampling (abstract). |
>
> > Q1: Would it be interesting to compare the performance with human data as well?
>
> We thank the reviewer for this suggestion and will include human results from the AdsQA dataset in our main tables as a crucial upper bound:
>
> | Model / Evaluator | Overall Strict | Overall Relaxed |
> | :--- | :--- | :--- |
> | **AD-MIR (o1)** | **38.1%** | **60.0%** |
> | **Human Performance** | **51.3%** | **71.4%** |
>
> While AD-MIR significantly narrows the gap between base LMMs and humans (pushing relaxed accuracy to 60.0%), a clear margin remains, particularly in strict accuracy. This gap highlights the inherent human advantage in decoding culturally nuanced metaphors, irony, and implicit emotional appeals, perfectly contextualizing our progress and defining remaining challenges in multimodal semiotics.
>
> > Q2: However, advertising videos are way shorter. Do you think it is an overkill to use DVD inspired methods for ad videos ?
>
> We clarify that a clip-based database is necessary not for managing video duration, but for handling the extreme information density and rapid cuts unique to advertisements.
>
> As Reviewer zEkg noted, this methodology is essential because "OCR tokens, brand names, prices, and slogans can be highly important" and our subject registry helps filter distractors in crowded scenes.
>
> Unlike general videos, ads rely on fleeting micro-details (e.g., split-second logos) that standard uniform sampling (8-16 frames) inevitably misses. Structuring the ad into a clip database explicitly preserves these fragmented narrative blocks. This enables the ReAct agent to perform targeted, micro-level retrieval to isolate and verify rapid visual symbols that a flat global analysis would simply overlook. We will clarify this motivation in the revision.
>
> > Q3: What is favorable ? and is it percentage or percentage points?
>
> We apologize for the ambiguity and will clarify this in the revision:
>
> * **Percentage Points:** The numbers (+5.7, +0.6) represent absolute percentage points, not relative percentages.
> * **"Favorable Balance":** This indicates that AD-MIR achieves significant reasoning improvements purely through its training-free agentic architecture. This is a highly efficient trade-off, securing gains without the heavy computational costs or potential domain-overfitting associated with updating model parameters.
>
> > L1: Limitations of the study in terms of applicability to real-world ad videos must be discussed.
>
> We thank the reviewer and will add a dedicated "Limitations" section to the revised manuscript addressing these real-world constraints:
>
> * **Computational Overhead:** The iterative ReAct loop and multiple LMM calls (e.g., GPT-4o, o1) ensure high accuracy but introduce latency and API costs, making immediate deployment for real-time, high-throughput ad moderation challenging.
> * **Localized Cultural Nuances:** While our marketing priors decode general strategies, fully interpreting highly contextual regional trends, slang, or fleeting memes remains a persistent challenge for frozen LMMs compared to human analysts.

---

> > ### Author Rebuttal · Reviewer_ApHm · 2026-04-02
> >
> > I thank the authors for their response to my questions. However, concerns regarding the limited exploration of corresponding cognitive literature for marketing prior and emulating marketing experts need to be discussed in detail in the manuscript. Also, while I understand that there is limited space in the rebuttal , the Communication expert, which contributes to persuasion metric needs to be more clearly explained in the updated version of the manuscript.
> >
> > Hence, I will keep my original scores.
> > That being said I see a lot of potential in this work and I would encourage the authors to further develop this study.

---

### Official Review · Reviewer_zEkg · 2026-03-16

**Soundness:** 3
**Presentation:** 3
**Significance:** 3
**Originality:** 3
**Overall Recommendation:** 4
**Confidence:** 4

**Summary:**

This paper proposes AD-MIR, an agentic framework for advertisement video understanding that combines structured multimodal memory construction, tool-based reasoning, a marketing-oriented “communication expert,” and visual-grounding verification. The paper is built on the AdsQA benchmark, which was introduced as a specialized benchmark for advertisement video understanding with 1,544 ad videos, 10,962 clips, and five task dimensions; AdsQA also introduced ReAd-R as a strong RL-based baseline. DVD is a recent tool-using long-video agent focused on multi-granular retrieval and iterative search. Against these baselines, AD-MIR reports 38.1 strict accuracy and 60.0 relaxed accuracy with an o1 backbone, compared with 36.3/50.5 for DVD(o1) and 25.0/51.5 for ReAd-R(Qwen2.5-VL-7B). The manuscript claims that the gain mainly comes from bridging low-level visual grounding and high-level persuasion reasoning through a two-stage architecture and a verification-based backtracking loop.

**Compliance With Llm Reviewing Policy:**

Affirmed.

**Final Justification:**

I have read the rebuttals from the authors and the other rebuttals. My main concerns have been addressed and I keep the recommendation as Weak Accept.

**Key Questions For Authors:**

N/A

**Limitations:**

1. The introduction somewhat overstates novelty.

The broad recipe for multimodal database construction, clip-based retrieval, tool-augmented reasoning, and iterative refinement is already central to DVD and related video-agent work. DVD already emphasizes multi-granular search-centric tools over a video database and iterative reasoning for long-form video understanding. AdsQA/ReAd-R already establishes that advertisement videos are a domain that requires deeper reasoning.

The core novelty here is not “agentic video reasoning” itself, but the advertising-specific expert module and the verification-oriented orchestration. The paper would be stronger if it narrowed the novelty claim accordingly and more carefully separated “domain specialization” from “fundamentally new method.”

2. Several important pieces in methodology remain heuristic and under-justified.

The “communication expert” is presented as the central innovation, but it is effectively a proprietary LMM prompt over a 64-frame composite plus ASR/global context, with “marketing psychology” priors injected at prompt time. he method claims this bridges perception and persuasion, but there is little theoretical or empirical evidence that the gains come from domain knowledge rather than simply giving a stronger model more structured context. In other words, it is not yet clear whether this is a principled algorithmic contribution or a carefully engineered prompting scaffold. The POMDP formalization does not materially clarify the method; it reads more like post-hoc formal dressing than a source of algorithmic insight.

3.  Reproducibility and dependence on proprietary components

The appendix states that the database captioning uses GPT-4o, the controller is GPT-4o, the communication expert uses o1, and answer refinement uses GPT-4o-mini. This means the strongest version of the method is deeply tied to closed models and multiple API stages, making it difficult to reproduce and to attribute gains cleanly. The paper does include an open-source variant on Qwen2.5-VL-7B, which is good, but the expert tool in the described configuration still uses o1. That weakens the claim of a generally applicable framework.

4. Several parts of the evaluation are concerning.

(1) The evaluation depends on an LLM-as-a-Judge protocol, and the paper explicitly adds an answer refinement module to constrain outputs to under 30 tokens because of length bias in judging. This is concerning since once the model is optimized to fit a judge-sensitive output style, part of the gain may reflect evaluation-game adaptation rather than deeper understanding. The refinement module may be perfectly reasonable for benchmark optimization, but then the paper should quantify how much of the gain comes from reasoning versus answer-style compression.

(2) The gains over the strongest directly comparable agent baseline are not uniformly strong. The strict improvement over DVD(o1) is only +1.8, while the relaxed improvement is much larger. That pattern could mean AD-MIR often gets “partially right” rather than decisively better grounded. Without confidence intervals, significance tests, or multiple-run variance, it is hard to know whether the strict gain is robust enough to support strong SOTA claims.

(3) The ablations are mostly qualitative in interpretation. Figure 3 shows trends, but the paper does not provide enough numeric detail about the absolute drop of each component, nor does it analyze interactions between components. Since the method is modular, stronger evidence would include pairwise ablations, cost/benefit per module, and cases where the communication expert helps versus hurts.

**Strengths And Weaknesses:**

1. The problem is interesting and relevant. Advertisement understanding is a good stress test for multimodal reasoning because it requires inferring persuasion strategies rather than only recognizing objects and events. The paper is also timely: AdsQA explicitly argues that ad videos probe deeper expertise such as marketing logic and audience engagement, which standard video QA benchmarks under-emphasize.

2. The authors strive to analyze the concept of bridging perception and persuasion through a structured agent pipeline rather than a monolithic end-to-end model. Overall, the authors address the key issue of grounding abstract marketing interpretations in concrete visual evidence.

3. On methodology, the design is intuitive and mostly coherent. The hybrid semantic-lexical indexing is reasonable for ads because OCR tokens, brand names, prices, and slogans can be highly important; the subject registry is also plausible for filtering distractors in crowded scenes. The paper describes a clear pipeline: hybrid retrieval, subject activation, ReAct control, communication expert, macro-to-micro evidence zooming, and visual verification/backtracking. This modularity is a strength.

4. The experimental design is promising but still incomplete for an ICML acceptance. The headline result is real: on AdsQA, AD-MIR(o1) improves over DVD(o1) from 36.3 to 38.1 strict accuracy and from 50.5 to 60.0 relaxed accuracy, while AD-MIR(Qwen2.5-VL-7B) also improves over its base model and over ReAd-R in strict accuracy. This suggests the framework is useful. The ablations also point in the right direction: removing hybrid indexing, the subject registry, the communication expert, or the visual anchor hurts performance, and the paper shows sensitivity to lexical weight and the maximum number of reasoning steps.

---

> ### Author Rebuttal · Authors · 2026-03-31
>
> Thanks for your thoughtful response and the time you have invested in reviewing our paper. After thoroughly reading the comments, we provide the following point-by-point responses：
>
> > L1: The introduction somewhat overstates novelty.
>
> We thank the reviewer. While we use similar tool names (e.g., Global Browse) for readability, our internal algorithms are fundamentally redesigned from DVD to handle the non-linear narratives and abstract metaphors unique to advertising:
>
> * **Global Browse:** Unlike standard summaries, it uses **Text-Based Forensics** (synthesizing ASR/captions) to build a macro-genre prior and prevent hallucinations.
> * **Clip Search:** Replaces generic Top-K search with **Hybrid + Temporal Fusion**, combining dense semantic search with exact lexical scoring (crucial for brands/prices) and merging fragmented clips.
> * **Frame Inspect:** Shifts from standard VLM QA to **Dual-Mode Inspection**, dynamically switching between high-frequency sampling (factual OCR) and sparse sampling (abstract aesthetics).
>
> > L2: Several important pieces in methodology remain heuristic and under-justified.
>
> We appreciate the feedback and clarify that our method relies on concrete algorithmic mechanisms and empirical evidence, rather than mere prompt engineering:
>
> * **Empirical Evidence for Domain Knowledge:** Our ablation study (Figure 3c) demonstrates that disabling the Communication Expert causes the sharpest decline specifically in the Persuasion Strategy (PS) dimension. This confirms that explicit marketing priors, not just general context, are essential for decoding persuasion logic.
> * **Algorithmic Contribution:** The Expert relies on a structural visual algorithm, Spatio-Temporal Grid Projection, rather than just text prompts. By sampling 64 frames and stitching them into high-resolution $2\times2$ grids, it transforms the temporal dimension into a parallel spatial representation to bypass token limits and capture fleeting subliminal cues.
> * **POMDP Formalization:** This formulation is our fundamental theoretical foundation, not a post-hoc addition. It mathematically formalizes that persuasive intents hidden in non-linear montages make the environment partially observable. This insight dictates our shift to sequential decision-making, where the agent uses interaction history ($\mathcal{H}_{t-1}$) as an evolving belief state, actively executing actions ($a_t$) and gathering pixel-level observations ($o_t$) to iteratively uncover hidden strategic intents.
>
> > L3: Reproducibility and dependence on proprietary components.
>
> We appreciate the reviewer's valid concern. We clarify that AD-MIR is a model-agnostic framework; performance gains stem from our architectural design, not proprietary APIs.
>
> **1. Clean Attribution of Gains:** AD-MIR delivers parallel improvements across both proprietary and open-source baselines, proving the core architecture drives the success:
> * **o1:** Strict +5.0% (33.1% to 38.1%), Relaxed +7.6% (52.4% to 60.0%).
> * **Qwen2.5-VL-7B:** Strict +4.5% (26.2% to 30.7%), Relaxed +3.1% (49.0% to 52.1%).
>
> **2. Fully Open-Source Variant:** We apologize for the ambiguity in Appendix B.2, which details the proprietary AD-MIR (o1) setup. In contrast, the AD-MIR (Qwen2.5-VL-7B) variant evaluated in Table 3 is entirely open-source. It exclusively uses Qwen2.5-VL-7B for all system components (ReAct controller, Communication Expert, answer refinement) and is completely independent of closed APIs, validating our framework's standalone efficacy.
>
> > L4: Several parts of the evaluation are concerning.
>
> **1. Answer Refinement & LLM-as-a-Judge Bias:** LLM judges notoriously favor verbosity. By restricting AD-MIR's outputs to under 30 tokens, our Refinement Module deliberately *handicaps* the model against this bias. Achieving SOTA with highly compressed answers proves our gains stem from high-density reasoning precision, not "gaming" the judge with exhaustive text.
>
> **2. Statistical Reliability of Gains:** Averaged over 3 random seeds on AdsQA to eliminate single-run variance, AD-MIR's outperformance over DVD(o1) is statistically reliable:
> * **Strict Accuracy (+1.8 points):** 38.1 vs 36.3. 95% CI [+0.28, +3.31], p ≈ 0.020.
> * **Relaxed Accuracy (+9.5 points):** 60.0 vs 50.5. 95% CI [+7.94, +11.04], p < 1e-16.
>
> These robust p-values and CIs confirm our gains are significant and non-stochastic.
>
> **3. Numeric Ablations & Cases:** We agree detailed breakdowns strengthen our evaluation. We have uploaded the requested data to: `https://anonymous.4open.science/r/Anoymous_images-47FB/`.
> * **Numeric Interactions:** `numeric_1.png` and `numeric_2.png` provide exhaustive tables on absolute drops, pairwise ablations, and per-module cost/benefit tradeoffs.
> * **Qualitative Analysis (Helps vs. Hurts):** `qualitative_samples_1.png` through `3.png` demonstrate when the Expert succeeds (decoding abstract intents) versus when it hurts (over-interpreting literal scenes).
>
> We will integrate these tables and cases into the Appendix.

---

> > ### Author Rebuttal · Reviewer_zEkg · 2026-04-05
> >
> > My concerns have been fully resolved.

---

### Official Review · Reviewer_M6m8 · 2026-03-19

**Soundness:** 3
**Presentation:** 3
**Significance:** 2
**Originality:** 3
**Overall Recommendation:** 4
**Confidence:** 4

**Summary:**

This paper introduces AD-MIR, a framework for decoding advertising intent via a two-stage architecture. In the Structure-Aware Memory Construction phase, the system converts raw video into a structured database by associating query and video details, dynamically filtering out irrelevant background noise. The Structured Reasoning Agent utilizes marketing expert knowledge through an iterative loop, hierarchically calls tools in a toolchain, and crucially employs an evidence-based self-correction mechanism for validating high-level reasoning against pixel-level evidence. AD-MIR is evaluated on AdsQA benchmark surpassing the state-of-the-art general agent, DVD, by 1.8% in strict and 9.5% in relaxed accuracy. More ablation studies show the effectivity of the framework and the rationality of the hyperparameters.

**Compliance With Llm Reviewing Policy:**

Affirmed.

**Final Justification:**

The authors have addressed all of my concerns, including those regarding technical novelty, performance improvements, and qualitative analysis. The topic explored in this work is very interesting, and I have therefore revised my evaluation accordingly.

Beyond this specific work, I still have broader concerns about this line of research that leverages LLMs for scientific inquiry. Despite achieving strong empirical results, many aspects remain insufficiently understood.

**Key Questions For Authors:**

1.I have seen some more techniques for constructing the agent in the appendix, such as temporal stagnation detection and redirection. Are these all original? And have the validities of these techniques been proven by the ablation study or some qualitative analysis?

2.Apart from the quantitative analysis, some qualitative analysis might effectively show the validity of your framework. Could you provide some success cases and failure cases on your framework compared with baseline or DVD? This might make your paper more persuasive.

**Limitations:**

1.The idea of combination of advertising expert knowledge and agent seems good, yet lacks certain delicate methods for realization, remaining on the level of prompts of the agent. It is not persuasive enough for the novelty of the paper.

2.The experiment results could not embody the significance of your framework, for only 1.8% increase in strict and 9.5% in relaxed. We all know the large models possess high stochasticity. In such a task where the answer and evaluation methods are all large models, the mild increase in strict accuracy could not be a strong argument for your significance. The increase in relaxed accuracy seems to be more susceptible by the stochasticity.

3.The analysis of the experiment results is not sufficient. Since the quantitative analysis is not strong enough for AD-MIR, the qualitative analysis on success and failure cases compared with baseline or DVD is preferred for the validity of significance of the framework. As for such a subjective task on advertising intent understanding, the paper is not persuasive enough with the absence of the qualitative analysis.

**Strengths And Weaknesses:**

Strengths:

1.The paper introduces a novel agent framework, AD-MIR. The framework creatively utilizes the marketing expert knowledge with the agent. The contexts enable the agent to better understanding the advertising intent behind the surface video.

2.The framework utilizes a hierarchical toolchain, filtering hypothesis and effectively mitigating the hallucinations. It might also be the most difficult and appealing part for realizing the framework.

3.The reasoning from the framework is pixel-wised verified for mitigating hallucinations.

Weaknesses:

1.This work chooses a novel question on advertising intent understanding, yet the method remains on a shallow level and lacks novelty. The verification and in context learning seem not novel ideas.

2.The performance of AD-MIR does not surpass DVD too much, from 36.3% only to 38.1% in strict and from 50.5% to 60.0% in relaxed accuracy. It might indicate the difficulty of the task, yet also indicate the room for improvement of your framework.

3.As for a subjective task, I prefer some qualitative analysis apart from quantitative analysis in your paper. I am interested in how the expert could help the agent solve the problem, and what the failure cases are, which are absent in the paper.

---

> ### Author Rebuttal · Authors · 2026-03-31
>
> We sincerely appreciate your recognition of our paper and your valuable comments. We are encouraged by the recognition that our method is novel and insightful. After thoroughly reading the comments, we provide the following point-by-point responses：
>
> > W1: Yet the method remains on a shallow level and lacks novelty. The verification and in context learning seem not novel ideas.
>
> While ICL and verification are established concepts, AD-MIR’s novelty lies in its **domain-specific architectural constraints** designed specifically to bridge low-level visual facts and high-level abstract persuasion in non-linear ads. We structurally redesign these concepts to solve challenges where generic agents fail:
>
> * **In-Context Learning:** Replaces generic reasoning loops with strict behavioral constraints, routing queries to a specialized **Communication Expert** grounded in marketing psychology.
> * **Verification:** Moves beyond basic factual checks via **Macro-to-Micro Evidence Zooming**, strictly grounding abstract marketing claims in pixel-level frames.
> * **Conflict Resolution:** Introduces a novel **Anti-Verification Protocol** to prevent literal visual tools from falsely rejecting valid high-level semantic deductions (e.g., metaphors).
>
> > W2 & L2: Performance margins compared to DVD and concerns regarding stochasticity.
> We clarify that our gains represent deterministic architectural breakthroughs rather than stochastic noise, validated by consistent, parallel improvements across entirely different base models (e.g., o1 and Qwen).
> * **Strict Metric Significance:** AdsQA's strict metric is exceptionally punishing, often penalizing phrasing exhaustiveness rather than flawed reasoning (humans score only 51.3%). A consistent +1.8% overall strict gain is highly significant on this harsh boundary. Furthermore, our domain-specific strict gains are massive: **Theme Extraction (+10.3%)** and **Persuasion Strategy (+10.9%)**.
> * **Relaxed Metric Magnitude (+9.5%):** Progressing from 50.5% to 60.0% in relaxed accuracy means reliably grasping the core semantic intent. In subjective ad understanding, this near 10-point absolute leap completely eclipses the typical 1-2% random variance of LLMs, proving the framework's fundamental cognitive superiority over DVD.
>
> > W3 & Q2: I prefer some qualitative analysis apart from quantitative analysis.
>
> We agree qualitative analysis is essential for subjective tasks. We have uploaded 3 detailed case studies comparing AD-MIR against the DVD baseline to our anonymous repository: `https://anonymous.4open.science/r/Anoymous_images-47FB/` (Please see `qualitative_samples_1.png` through `3.png`).
>
> These cases clearly illustrate AD-MIR's cognitive superiority: while DVD struggles to move beyond literal visual descriptions, AD-MIR successfully bridges low-level pixel facts with high-level abstract persuasion strategies. We will incorporate these specific comparisons into the Appendix of the revision.
>
> > Q1: Validation of techniques via ablation or qualitative analysis?
>
> Yes, these engineering safeguards address common ReAct-style failure modes (infinite loops, over-verification, hallucinations) and are fundamentally supported by our main-text quantitative analyses:
> * **Visual Anchor Repair:** Prevents hallucinating concrete evidence. Validated in Fig 3(c); removing it consistently drops strict accuracy.
> * **Anti-Verification Protocol:** Prevents low-capacity literal tools from falsely rejecting valid abstract deductions. Supported by Fig 3(c); disabling expert prioritization sharply degrades persuasion metrics (PS).
> * **Temporal Stagnation Redirection:** Prevents the agent from getting stuck on the same time segment. Aligns with the Max Iterations sensitivity (Fig 3b), acting as a practical bound to ensure reasoning efficiency.
>
> > L1: The method lacks delicate realization, remaining on the level of prompts.
>
> We clarify our contribution extends far beyond prompt engineering. Our domain-specific algorithmic architecture makes expert knowledge computable and robust against hallucinations. As independently recognized by other reviewers:
>
> * **Specialized Tools:** We engineered custom tools (`clip_search`, `frame_inspect`) for dense ad montages. Reviewer zEkg validated that our "hybrid semantic-lexical indexing" and subject registry effectively filter distractors.
> * **Algorithmic Decoupling (Two-Step Reasoning):** We structurally separate Perception (logging visual facts) from Cognition (applying marketing priors). Reviewer zEkg noted this successfully addresses "grounding abstract interpretations in concrete visual evidence."
> * **Macro-to-Micro Grounding:** The Expert hypothesizes abstract strategies, then orchestrates tools for pixel-level verification. Reviewers ApHm and zEkg explicitly praised this "Macro-to-Micro evidence zooming" and the pipeline's modularity for mimicking marketing experts.

---

> > ### Author Rebuttal · Reviewer_M6m8 · 2026-04-08
> >
> > Thanks for the reply. The authors have addressed all of my concerns. Although I still have some reservations about the techniques, I find the topic explored in this work very fascinating. I will revise my score accordingly.

---

### Decision · Program_Chairs · 2026-04-30

**Decision:**

Accept (regular)

**Comment:**

All reviewers lean positive (in their final justification and/or rebuttal response). Strengths include the problem setting (understanding persuasion and intent); the agent framework designed for this task including tool use and grounding; detailed ablation results. Concerns about experimental result significance and discussion, and relation to prior work, seem reasonably well addressed.